# *Obox4* promotes zygotic genome activation upon loss of *Dux*

**Youjia Guo[1†], Tomohiro Kitano[1†], Kimiko Inoue[2,3†], Kensaku Murano[1], Michiko Hirose[4], Ten D Li[1], Akihiko Sakashita[1,3], Hirotsugu Ishizu[1], Narumi Ogonuki[2], Shogo Matoba[2], Masayuki Sato[1], Atsuo Ogura[2,3]\*, Haruhiko Siomi[1,4]\***

[1]Department of Molecular Biology, Keio University School of Medicine, Tokyo, Japan; [2]Bioresource Engineering Division, Bioresource Center, RIKEN, Tsukuba, Japan; [3]Graduate School of Life and Environmental Sciences, University of Tsukuba, Tsukuba, Japan; [4]Human Biology Microbiome Quantum Research Center (WPI-Bio2Q), Keio University, Tokyo, Japan

**Abstract** Once fertilized, mouse zygotes rapidly proceed to zygotic genome activation (ZGA), during which long terminal repeats (LTRs) of murine endogenous retroviruses with leucine tRNA primer (MERVL) are activated by a conserved homeodomain-containing transcription factor, DUX. However, *Dux*-knockout embryos produce fertile mice, suggesting that ZGA is redundantly driven by an unknown factor(s). Here, we present multiple lines of evidence that the multicopy homeobox gene, *Obox4*, encodes a transcription factor that is highly expressed in mouse two-cell embryos and redundantly drives ZGA. Genome-wide profiling revealed that OBOX4 specifically binds and activates MERVL LTRs as well as a subset of murine endogenous retroviruses with lysine tRNA primer (MERVK) LTRs. Depletion of *Obox4* is tolerated by embryogenesis, whereas concomitant *Obox-4/Dux* depletion markedly compromises embryonic development. Our study identified OBOX4 as a transcription factor that provides genetic redundancy to preimplantation development.

**\*For correspondence:**
ogura@rtc.riken.go.jp (AO);
awa403@keio.jp (HS)

[†]These authors contributed equally to this work

**Competing interest:** The authors declare that no competing interests exist.

## Editor's evaluation

This study presents an important finding that Obox4 and Dux act redundantly in regulating zygotic genome activation in mice. The evidence supporting the claims of the authors is convincing. The work will be of interest to researchers interested in early embryo development and epigenetic reprogramming.

## Introduction

The mechanism by which zygotes acquire totipotency is a major question in developmental biology. Following fertilization, the zygote must build a de novo conceptus with a transcriptionally quiescent genome. The genome rapidly undergoes zygotic genome activation (ZGA), during which epigenetic reprogramming and expression of nascent transcripts enforce the replacement of parental infrastructures by their zygotic counterparts (*Lee et al., 2014*). Completion of metazoan ZGA results in blastomeres, a collection of cells that are totipotent enough to reflect their potential to individually produce both embryos and extraembryonic appendages (*Tarkowski, 1959a*; *Tarkowski, 1959b*). Totipotency lasts until the first cell fate decision takes place, following which it is remolded into pluripotency in cells destined for the embryonic lineage (*Suwińska et al., 2008*), to which embryonic stem cells (ESCs) and induced pluripotent stem cells (iPSCs) correspond. In placental mammals, ZGA is characterized by the massive reactivation of transposable elements (TEs) and epigenome remodeling, predominantly

implemented by a collection of ZGA genes that have adapted long terminal repeats (LTRs) of endogenous retroviruses (ERVs) as stage-specific *cis*-regulatory elements (*Svoboda et al., 2004*).

Homeodomains are DNA-binding amino acid motifs encoded by a class of conserved genomic sequences termed homeoboxes (*McGinnis et al., 1984b*; *Carrasco et al., 1984*; *McGinnis et al., 1984a*). Homeodomains are prevalent transcription factors that regulate development because of their ability to bind chromatin in a sequence-specific manner (*Scott and Weiner, 1984*; *Gehring and Hiromi, 1986*; *Gehring, 1987*). Among all homeobox-containing genes, Paired-like (PRD-like) homeobox class genes are particularly associated with ZGA (*Lewin et al., 2021*). Studies have shown that the PRD-like homeobox gene, double homeobox (*Dux*), plays an important role in preimplantation embryogenesis (*Töhönen et al., 2015*; *Madissoon et al., 2016*). *Dux* is conserved throughout placentalia and is highly expressed during ZGA. In humans and mice, DUX specifically binds and activates endogenous retroviruses with leucine tRNA primer (ERVL) LTR-derived promoters, resulting in the expression of downstream ZGA genes (*De Iaco et al., 2017*; *Hendrickson et al., 2017*; *Whiddon et al., 2017*). *Dux* expression induces a two-cell-embryo-like (2C-like) state in mouse embryonic stem cells (mESCs) (*De Iaco et al., 2017*; *Hendrickson et al., 2017*; *Whiddon et al., 2017*), whereas *Dux*-knockout (KO) undermines mouse blastocyst formation (*De Iaco et al., 2017*). However, subsequent studies have revealed that mouse embryogenesis is compatible with the loss of *Dux*, suggesting a redundant pathway of ZGA that is controlled by uncharacterized transcription factor(s) (*Chen and Zhang, 2019*; *Guo et al., 2019*; *De Iaco et al., 2020*; *Bosnakovski et al., 2021*).

In this study, we sought to identify the redundant transcription factor that drives ZGA in the absence of *Dux*. We discovered that the multicopy homeobox gene, oocyte-specific homeobox 4 (*Obox4*), promotes *Dux*-less ZGA. *Obox4* is abundantly expressed in mouse 2C-embryos, 2C-like mESCs, and totipotent blastomere-like cells (TBLCs) (*Shen et al., 2021*). Mechanistically, OBOX4 promotes ZGA by binding to the LTRs of murine endogenous retroviruses with leucine tRNA primer (MERVL) and murine endogenous retroviruses with lysine tRNA primer (MERVK), and thereby affecting the deposition of active epigenetic modifications. Concomitant, but not respective, depletion of *Obox4* and *Dux* severely compromises ZGA and preimplantation development. Taken together, our findings substantiate a preimplantation development model, in which the ZGA is redundantly promoted by *Dux* and *Obox4*.

## Results

### Expression profiling identifies transcription factor candidates

Genetic redundancy is commonly observed among genes that share sequence homology (*Wagner, 1996*). Because *Dux* is a homeobox gene, we speculated that its redundant factors are homeobox genes. We examined the expression profiles of all mouse homeobox genes during preimplantation embryogenesis using published single-cell RNA-seq (scRNA-seq) data (*Deng et al., 2014*). Clustering analysis defined a collection of 45 homeobox genes, whose transcript levels reached a maximum before the late 2C stage (*Figure 1A*). We then sought to identify bona fide ZGA factors from this collection by looking for genes whose transcripts were of zygotic origin, instead of ones that were parentally inherited. TBLCs have recently been established as an in vitro model of 2C-embryos, which are derived from splicing-inhibited mESCs, and hence, are free of parentally inherited transcripts (*Shen et al., 2021*). Analysis of published RNA sequencing (RNA-seq) data showed that the expression of four homeobox genes, *Duxf3*, *Emx2*, *Hoxd13*, and *Obox4*, was 10 times higher in TBLCs than in mESCs (*Figure 1B*). We examined whether the expression of these genes was affected in the *Dux* knockout embryos. Using published RNA-seq data (*Chen and Zhang, 2019*; *De Iaco et al., 2020*), we found that only the expression of the multicopy gene *Obox4* was partially affected in *Dux* knockout embryos (*Figure 1C*).

### *Obox4* activates 2C-genes and TEs in mESCs

Before examining whether the candidates were crucial for embryogenesis, we first confirmed the presence of endogenous OBOX4 protein as the *Obox4* loci are marked as pseudogenes in the current genome annotation GRCm39 (*Yates et al., 2020*). We generated mouse anti-OBOX4 monoclonal antibodies (*Figure 1—figure supplement 1*). Immunofluorescence staining using these antibodies confirmed that endogenous OBOX4 was expressed in zygotes and highly abundant in 2C-embryos and

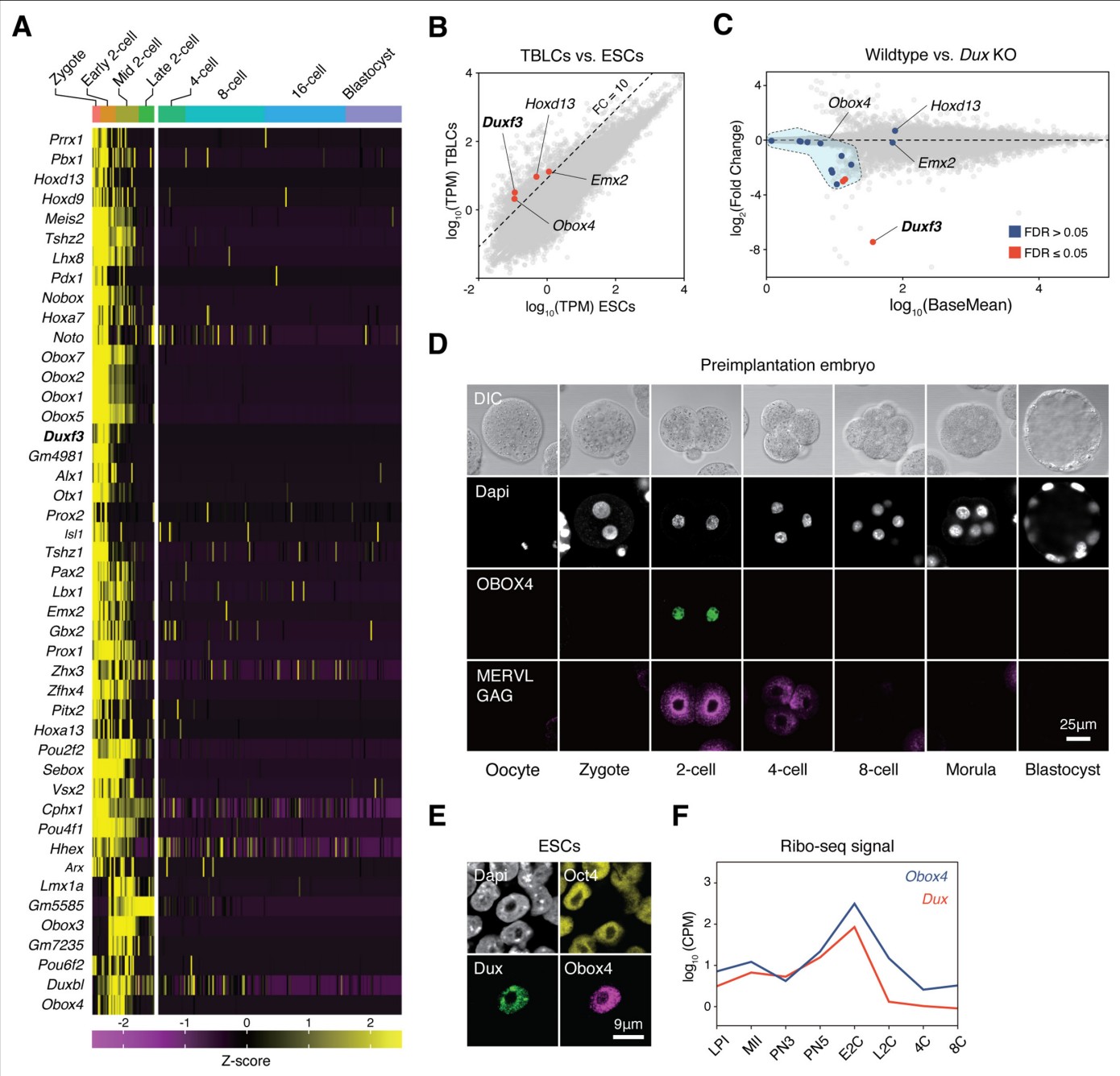

**Figure 1.** OBOX4 is expressed during zygotic genome activation (ZGA). (**A**) Mouse homeobox genes that are specifically expressed during ZGA. The genes were identified by means of statistically determined *k*-means clustering based on their expression in preimplantation embryos. *Dux* is shown in a bold italic font. (**B**) Scatterplot showing per-gene normalized read counts in mouse totipotent totipotent blastomere-like cells (TBLCs) versus mouse embryonic stem cells (mESCs). Genes with more than tenfold normalized read counts (FC = 10) have been highlighted. *Dux* is shown in a bold italic font. (**C**) MA plot displaying gene expression in *Dux* knockout 2C-embryos versus WT 2C-embryos. Loss of *Dux* partially downregulates the multicopy homeobox gene *Obox4*. (**D**) Immunofluorescence staining of OBOX4 and MERVL GAG at different preimplantation embryo stages. (**E**) Immunofluorescence staining of DUX, OBOX4, and OCT4 in 2C-like mESCs. (**F**) Translation profile of DUX and OBOX4 during preimplantation characterized by Ribo-seq signal.

The online version of this article includes the following figure supplement(s) for figure 1:

**Figure supplement 1.** Production of anti-OBOX4 monoclonal antibodies.

2C-like mESCs (*Figure 1D and E*), which is consistent with translatome profiles captured by Ribo-seq that detects OBOX4 translation at the 2C stage (*Xiong et al., 2022*; *Figure 1F*). Mouse ZGA is characterized by the surging activity of genes that specifically express at the 2C stage (2C-gene), many of which have co-opted MT2_Mm, the LTR of MERVL, as stage-specific promoters (*Macfarlan et al., 2011*). We then sought to examine whether the candidates had potential to promote ZGA by activating MERVL and 2C-genes (*Figure 2—figure supplement 1A*; *Supplementary file 1*). We constructed a 2C::tdTomato reporter mESC line with transgenic tdTomato red fluorescence protein driven by the MERVL 5'-LTR (*Macfarlan et al., 2012*; *Figure 2A*). Candidate genes were cloned and overexpressed in the reporter cell line (*Figure 2—figure supplement 2B and C*). Interestingly, ectopic expression of *Obox4* markedly induced tdTomato expression, similar to that induced by *Dux* (*Figure 2B*, *Figure 2—figure supplement 2A and B*). To characterize the impact of *Obox4* expression on 2C-genes, we established an mESC stable line bearing an inducible *Obox4* transgene (*Figure 2C*). Differentially expressed gene analysis revealed that *Obox4* induction led to transcriptome changes in mESCs, characterized by upregulation of 2C-genes and TEs that were highly expressed in early-to-middle 2C-embryos (*Wu et al., 2016*; *Figure 2D and E*), naturally occurring 2C-like mESCs (*Zhu et al., 2021*), and *Dux*-induced 2C-like mESCs (*Hendrickson et al., 2017*; *Figure 2F*; *Supplementary file 2*). A substantial fraction (159/164) of *Obox4*-induced 2C-genes were also induced by *Dux* (*Figure 2G*). While the induction effect of *Obox4* is milder compared with *Dux* when considering the number and fold-upregulation of 2C-genes, transcriptomic perturbances introduced by *Dux* and *Obox4* are highly correlated (*Figure 2H*). Interestingly, *Dux* and *Obox4* were mutually inductive, where one promoted expression of the other. These results suggest that *Obox4* is an inducer of 2C-like genes and potentially a redundant factor of *Dux*.

## Obox4 binds to 2C-gene promoters and LTR elements in mESCs

*Obox4* contains a homeobox, activates 2C-genes and TEs, and upregulates *Dux*-induced genes, which prompted us to examine whether OBOX4 is a transcription factor that activates 2C-gene-associated loci through direct binding with *cis*-regulatory elements. Cleavage under targets and release using nuclease (CUT&RUN) leverages antibody-targeted cleavage of proximal DNA to identify the binding sites of DNA-associated proteins (*Skene and Henikoff, 2017*). To examine this technique, we performed CUT&RUN against triple FLAG-tagged DUX (3×FLAG-DUX) expressed in mESCs using a high-affinity anti-FLAG antibody (*Sasaki et al., 2012*) and confirmed that the DUX binding pattern revealed by CUT&RUN recapitulated the published HA-tagged DUX ChIP data (*Hendrickson et al., 2017*; *Figure 3—figure supplement 1A and B*). We then proceeded with characterizing the genomic footprint of OBOX4 by performing CUT&RUN against 3×FLAG-OBOX4 expressed in mESCs and discovered ~24,000 peaks, among which 39.5% were located in the gene promoter regions that covered 26.8% (273/1019) of the 2C-genes (*Figure 3A*). De novo motif discovery using the top 500 CUT&RUN signal peaks predicted CTGGGATYWRMR as the top OBOX4 binding motif, which is enriched in the promoter regions of 2C-genes (*Figure 3B*). Collectively, OBOX4 and DUX targeted 48.1% (490/1019) of the 2C-genes with a considerable overlap (36.3% for OBOX4 and 31.3% for DUX), which covered many important ZGA genes, including *Dppa2*, *Sp110*, and *Zscan4d* (*Figure 3C and D*; *Supplementary file 3*). The overlap was substantiated by the observation that MERVL LTR MT2_Mm was among the top 10 LTR targets of both OBOX4 and DUX, based on loci coverage (*Figure 3E and F*; *Supplementary file 4*). Notably, while DUX strongly prefers MT2_Mm, OBOX4 binding was biased toward MERVK LTRs, namely RLTR9 and RLTR13 elements (*Figure 3G*), which also demonstrated a 2C-specific expression profile (*Figure 3—figure supplement 1C and D*). To examine whether these bindings were functional in the absence of *Dux*, we generated Obox4/Dux single and double knockout mESC lines, in which 14 of 15 protein-coding *Obox4* copies were removed from the genome (*Figure 3—figure supplements 2 and 3*). tdTomato reporters driven by MT2_Mm and RLTR13B2 were co-transfected with *Dux* and *Obox4* into *Obox4/Dux* double knockout mESCs (*Figure 3H*). Analysis of the tdTomato-positive population showed that *Obox4* activated both RLTR13B2 and MT2_Mm, whereas *Dux* only activated MT2_Mm (*Figure 3I*). These observations demonstrated that OBOX4 binds and activates a subset of DUX targets in mESCs and redundantly drives their expression in the absence of DUX.

## Concomitant loss of *Obox4* and *Dux* impairs preimplantation development

We then sought to determine whether *Obox4* is functionally required for preimplantation development, particularly in the absence of *Dux*. Transient depletion of DUX has previously confused the field

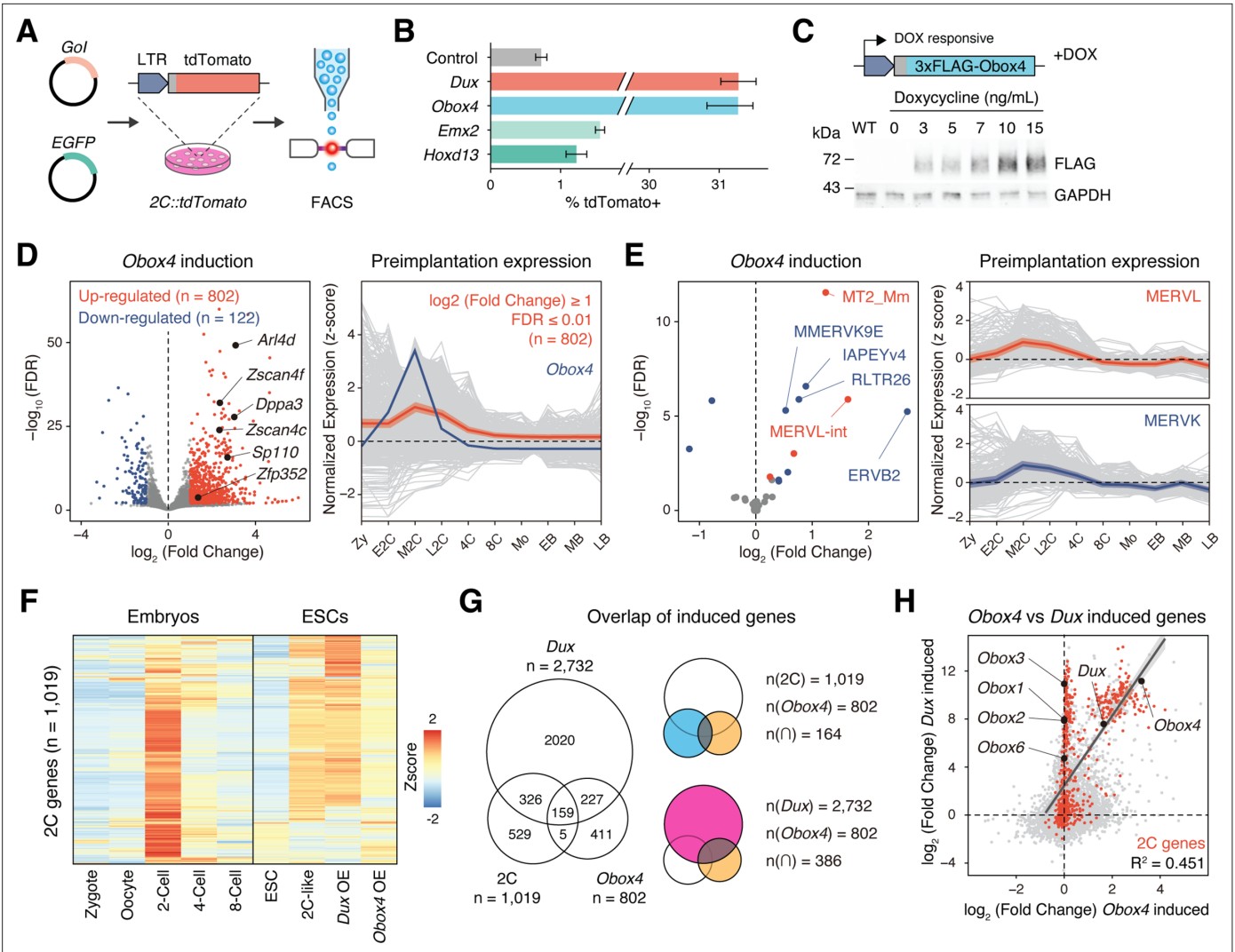

**Figure 2.** *Obox4* and *Dux* induce 2C-gene expression in mouse embryonic stem cells (mESCs). (**A**) Diagram of the 2C::tdTomato reporter assay. mESCs bearing the tdTomato expression cassette under the control of the MERVL long terminal repeat (LTR) promoter showed red fluorescence upon entering the 2C-like state. The expression of the transcription factor increased the 2C-like population, as detected using FACS. An EGFP expression plasmid was co-transfected with a gene of interest to normalize the transfection efficiency. (**B**) Boxplot showing normalized 2C-like cell percentage in 2C::tdTomato reporter mESCs overexpressing candidate pioneer factors. *Dux* and *Obox4* potently induced a 2C-like state. n = 3 biological replicates. Error bars indicate standard deviations. (**C**) Upper panel: schematic of *Obox4*-inducible cell line construction. Lower panel: western blot showing OBOX4 level upon induction by different concentrations of doxycycline. Expression of OBOX4 was carried out in a dose-dependent manner. (**D**) Left panel: volcano plot of differentially expressed genes (DEGs) in mESCs with *Obox4* induction for 48 hr. Representative 2C-genes are labeled with gene symbols. Right panel: expression profile of genes upregulated by *Obox4* during embryogenesis. n = 3 biological replicates. (**E**) Left panel: volcano plot of differentially expressed transposable elements in mESCs with *Obox4* induction for 48 hr. MERVL and MERVK elements were highlighted. Right panel: expression profile of MERVL and MERVK elements during preimplantation embryogenesis. n = 3 biological replicates. (**F**) Heatmaps of the expression of 2C-genes in preimplantation embryos, naturally occurred 2C-like mESCs, and induced 2C-like mESCs. (**G**) Venn diagram showing overlap of 2C-genes with genes induced by ectopic expression of *Dux* and *Obox4* in mESCs. (**H**) Scatterplot showing per-gene expression changes in *Dux*-induced versus *Obox4*-induced mESCs. 2C-genes are highlighted in red.

The online version of this article includes the following source data and figure supplement(s) for figure 2:

**Source data 1.** Original and uncropped blot for *Figure 2C*.

**Figure supplement 1.** *k*-means clustering of reimplantation gene expression and MERVL induction by ectopic *Obox4* and *Dux* expression.

**Figure supplement 1—source data 1.** Original and uncropped blot for *Figure 2—figure supplement 1B*.

**Figure supplement 2.** Ectopic expression of Obox4 and Dux activates MERVL reporter.

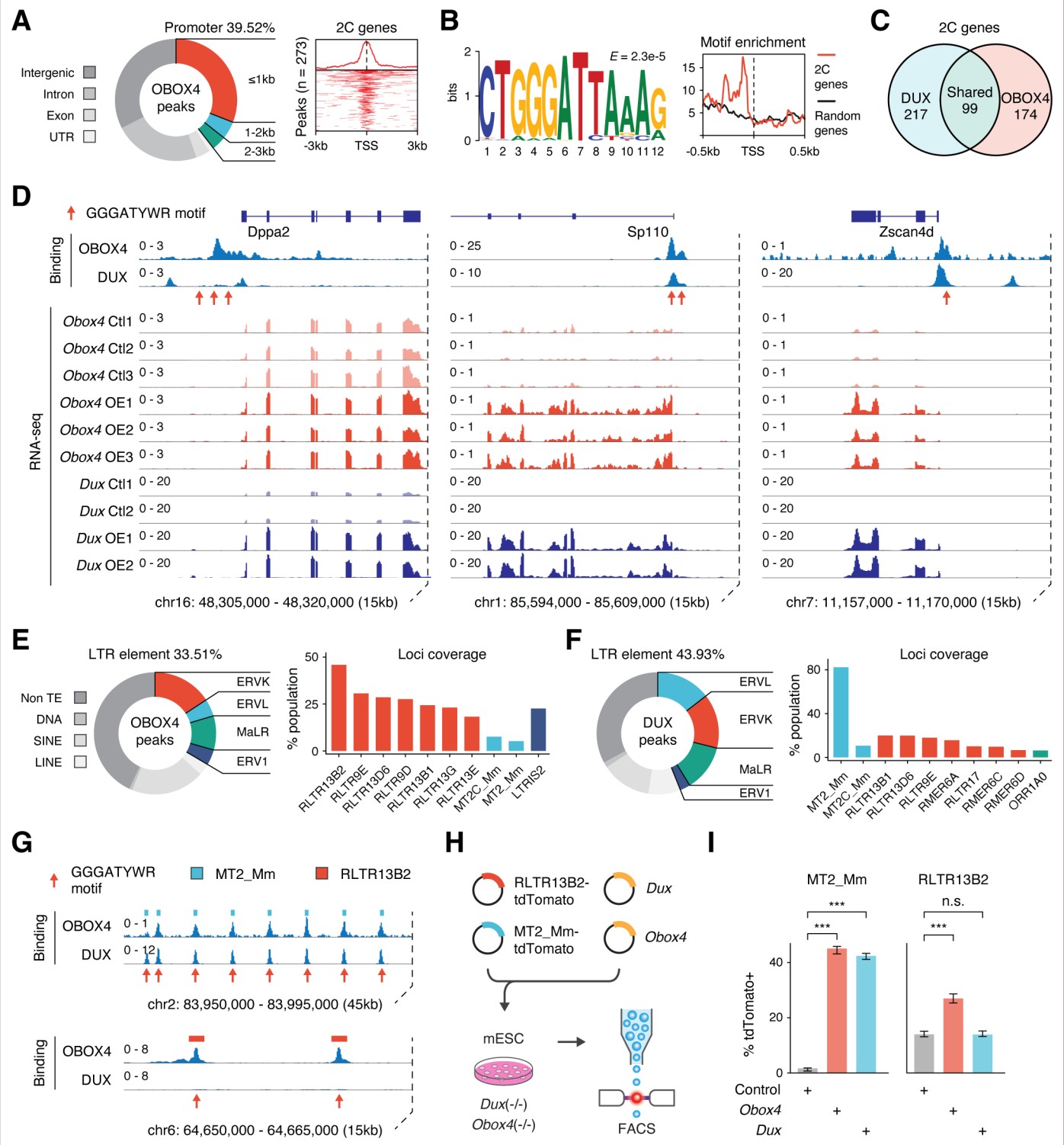

**Figure 3.** OBOX4 binds and activates 2C-specific long terminal repeat (LTR) elements. (**A**) Left panel: pie chart displaying proportions of annotated genomic regions of the OBOX4 binding sites. Right panel: heatmap showing the OBOX4 binding site distribution near 2C-gene promoters. (**B**) Left panel: the predicted OBOX4 binding motif using the top 500 cleavage under targets and release using nuclease (CUT&RUN) peaks. Right panel: histogram showing the distribution of the predicted OBOX4 binding motif near 2C and random gene promoters. (**C**) Venn diagram showing the distinct and overlapping 2C-genes targeted by DUX and OBOX4. (**D**) Representative genomic track showing DUX and OBOX4 binding sites at the *Dppa2, Sp110,* and *Zscan4d* loci and their expression levels in *Obox4* and *Dux*-induced mouse embryonic stem cells (mESCs). Read counts were CPM normalized. The OBOX4 binding sites overlapped with those of DUX. *Dppa2, Sp110*, and *Zscan4d* expression was upregulated upon *Obox4* and *Dux*

*Figure 3 continued on next page*

*Figure 3 continued*

induction. (**E**) Left panel: pie- hart displaying proportions of annotated transposable elements (TEs) of the OBOX4 binding sites. Right panel: bar plot showing the top 10 OBOX4 covered LTR elements. (**F**) Left panel: pie chart displaying proportions of annotated TEs of the DUX binding sites. Right panel: bar plot showing the top 10 DUX covered LTR elements. (**G**) Representative genomic track showing DUX and OBOX4 binding sites at MT2_Mm and RLTR13B2. Read counts were CPM normalized. The OBOX4 binding sites overlapped with those of DUX at MT2_Mm loci, while RLTR13B2 was uniquely bound by OBOX4. (**H**) Schematic design of the LTR::tdTomato reporter assay. Plasmids bearing a tdTomato ORF downstream of MT2_Mm or RLTR13B2 were co-transfected with *Dux* or *Obox4* expression plasmids. Activation of LTR elements resulted in an increased red fluorescence-positive mESC population, which was measured using FACS. EGFP expression plasmid was co-transfected into the culture, following which the green fluorescence-positive population was measured, to normalize the transfection efficiency. (**I**) Bar plots showing the percentage of red fluorescence-positive mESCs upon expression of *Dux* or *Obox4*. Both *Dux* and *Obox4* activated MT2_Mm, whereas only *Obox4* activated RLTR13B2.

The online version of this article includes the following source data and figure supplement(s) for figure 3:

**Figure supplement 1.** CUT&RUN captures DNA binding profiles of DUX and OBOX4.

**Figure supplement 2.** Knockout strategy of *Obox4* and *Dux* clusters.

**Figure supplement 3.** Validation of *Obox4* and *Dux* knockout mouse embryonic stem cell lines.

**Figure supplement 3—source data 1.** Original and uncropped gels for *Figure 3—figure supplement 3A*.

**Figure supplement 3—source data 2.** Original and uncropped gels for *Figure 3—figure supplement 3B*.

in that it caused an embryonic phenotype while subsequent *Dux* knockout females were shown to be fertile (*De Iaco et al., 2017*; *Chen and Zhang, 2019*; *Guo et al., 2019*; *De Iaco et al., 2020*; *Bosna-kovski et al., 2021*). Therefore, it is critical to examine the functional requirement of OBOX4 and DUX in genetic knockout models. Somatic cell nuclear transfer (SCNT) is a technique to create embryos by transferring nuclei of somatic cells into enucleated oocytes (*Figure 4A*), which recapitulates ZGA (*Markoulaki et al., 2008*). When the knockout mESCs were subjected to SCNT as nuclear donors, 54.4% of WT mESC-derived embryos developed to the blastocyst stage 4 days post nuclear transfer (dpt). *Obox4* and *Dux* single knockout donors led to reduced blastocyst formation rate at 38.5 and 39.7%, respectively, whereas double knockout resulted in an additive effect that led to a significantly lower blastocyst rate of 29.3% (*Figure 4B*). Notably, all blastocysts derived from double knockout mESCs were of low quality, judging from severe morphological abnormalities (*Figure 4C*).

As an alternative genetic knockout model, we generated mice bearing *Dux* and *Obox4* knockout alleles by direct CRISPR-Cas9 editing in embryos (*Figure 4D*, *Figure 3—figure supplement 2A and B*). Among 39 *Obox4* copies, there are 15 copies that maintain an intact open-reading frame (ORF) for full-length OBOX4 (*Figure 3—figure supplement 2C*). While 14 of the 15 intact ORFs form a tightly packed cluster (*Obox4* cluster), a solo ORF (*Obox4-ps33*) remains distant from the *Obox4* cluster and is interspersed with *Obox1/2/3*, a collection of *Obox* family members that are critical for ZGA (*Ji et al., 2023*). To minimize collateral genetic toxicity and interference to the experimental result caused by the removal of other *Obox* members, only the *Obox4* cluster was knocked out and the solo ORF was retained (*Figure 3—figure supplement 2D*, *Figure 4—figure supplement 1*). The observed *Obox4*KO frequency (7/29) in the progenies of *Obox4*Het × *Obox4*Het crossing is consistent with the expected 25% Mendelian ratio (*Figure 4E*, *Figure 4—figure supplement 2*; *Supplementary file 5*). That *Obox4*KO mice developed to adulthood without discernable abnormalities and are fertile when intercrossed suggest that development is compatible with loss of *Obox4* (*Figure 4F and G*). We next attempted to produce *Dux*/*Obox4* double knockout (*Dux*KO/*Obox4*KO) mice to examine whether concomitant loss of *Dux* and *Obox4* compromises embryogenesis. We genotyped 54 pups from three litters of *Dux*KO/*Obox4*Het × *Dux*KO/*Obox4*Het and six litters of *Dux*KO/*Obox4*Het × *Dux*Het/*O-box4*Het mating pairs (*Figure 4—figure supplement 3*; *Supplementary file 5*). One *Obox4*KO/*Dux*KO pup was present, and the frequency of *Obox4*KO/*Dux*KO and *Obox4*KO/*Dux*Het were significantly lower than the expected 25% Mendelian ratio (*Figure 4H*). Development monitoring and genotyping of embryos produced by *Dux*KO/*Obox4*Het × *Dux*Het/*Obox4*Het mating pairs at 4.5 days post coitum (dpc) revealed that *Dux*KO/*Obox4*KO was under-represented in blastocysts while over-represented in two-cell arrest and degenerated embryos (*Figure 4I*, *Figure 4—figure supplement 4A–C*; *Supplementary file 6*). Despite repeated attempts, various crossing strategies failed to produce *Obox4*KO/*Dux*KO mating pairs that could be used to produce large number of *Obox4*KO/*Dux*KO embryos required for transcriptome analysis. As an alternative, we performed single blastomere genotyping and RNA-seq of 2C embryos produced by *Dux*KO/*Obox4*Het crossings (*Figure 4—figure supplement 4D and E*),

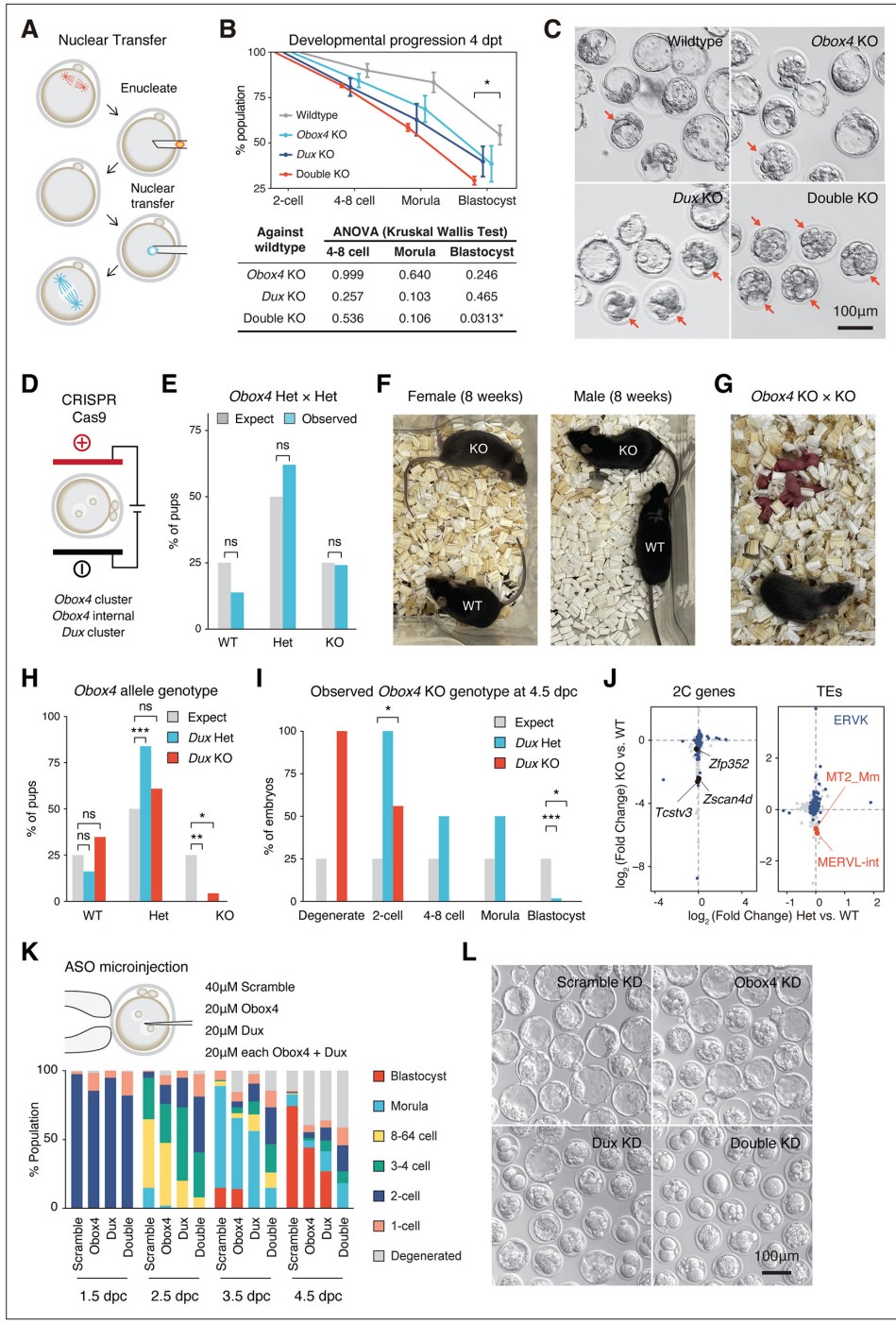

**Figure 4.** Concomitant loss of OBOX4 and DUX severely hinders zygotic genome activation (ZGA). (**A**) Schematic representation of the somatic cell nuclear transfer (SCNT) experiment. The nuclei of knockout mouse embryonic stem cells (mESCs) were transferred into enucleated oocytes to generate zygotes with knockout genotype. (**B**) Upper panel: percent SCNT embryos developed to different stages at 4 days post nuclear transfer (dpt). Lower panel: p-value of non-parametric ANOVA among different genotypes and stages compared to wildtype SCNT embryos. Three independent experiments were conducted, with 150–200 embryos per condition. (**C**) Representative picture of SCNT embryos at 4 dpt. Morphologically abnormal blastocysts are highlighted. Blastocysts generated by double knockout mESCs were severely defective. (**D**) Schematic representation of CRISPR-Cas9-mediated *Dux* and *Obox4* knockout mouse production. In vitro fertilized mouse zygotes were electroporated with pre-assembled CRISPR-Cas9 complex targeting *Dux* and *Obox4* loci. (**E**) Bar plot showing genotype percentage of the pups delivered by *Obox4*^Het intercrosses. Four litters delivered 29 pups, litter size 7.25

*Figure 4 continued on next page*

*Figure 4 continued*

± 1.26. ns p-value=0.9146, chi-square goodness-of-fit test. (**F**) Representative photos of *Obox4*<sup>KO</sup> and WT adult mice analyzed in (**E**). (**G**) Photo of *Obox4*<sup>KO</sup> intercross litter with live pups. (**H**) Bar plot showing genotype of *Obox4* allele in the pups delivered by crossing of *Dux*<sup>KO</sup>/*Obox4*<sup>Het</sup> × *Dux*<sup>KO</sup>/*Obox4*<sup>Het</sup> or *Dux*<sup>KO</sup>/*Obox4*<sup>Het</sup> × *Dux*<sup>Het</sup>/*Obox4*<sup>Het</sup>. Nine litters delivered 54 pups, *Dux* heterozygous and knockout allele were present in 31 and 23 pups, respectively. **p-value=0.001306; *p-value=0.02218; chi-square goodness-of-fit test. (**I**) Bar plot showing observed percentages of different preimplantation stage embryos bearing *Obox4* KO allele with *Dux* heterozygous or knockout allele at 4.5 days post coitum (dpc). Among the total 94 embryos assessed, 2 degenerated, 10 two-cell arrest, 2 4–8-cell arrest, 4 morula arrest embryos were observed at 4.5 dpc, whereas 76 developed to blastocyst. ***p-value=3.564 × 10$^{-5}$; for two-cell; * p-value=0.04348; for blastocyst *p-value=0.02549; chi-square goodness-of-fit test. (**J**) Scatterplot showing expression log$_2$ fold changes of genes (left panel) and transposable elements (TEs) (right panel) in *Dux*<sup>KO</sup>/*Obox4*<sup>KO</sup> and *Dux*<sup>KO</sup>/*Obox4*<sup>Het</sup> versus *Dux*<sup>KO</sup>/*Obox4*<sup>WT</sup> 2C embryos. 2C-genes and ERVK elements targeted by OBOX4 are highlighted in blue. MT2_Mm and MERVL-int are labeled. In total, 79 and 5 2C-genes were downregulated in *Dux*<sup>KO</sup>/*Obox4*<sup>KO</sup> and *Dux*<sup>KO</sup>/*Obox4*<sup>Het</sup> 2C embryos, respectively. (**K**) Upper panel: schematic representation of *Dux* and *Obox4* knockdown experiments in preimplantation embryos. Male pronuclei of zygotes were microinjected with antisense oligonucleotide (ASO) targeting *Dux* or *Obox4* transcripts. Lower panel: the percentages of embryonic stages observed at 1.5 dpc, 2.5 dpc, 3.5 dpc, and 4.5 dpc. (**L**) Representative picture of KD embryos at 4.5 dpc. No blastocyst was observed among double ASO knockdown embryos.

The online version of this article includes the following source data and figure supplement(s) for figure 4:

**Figure supplement 1.** Genotyping of *Obox4* and *Dux* knockout alleles in the F$_1$ mice.

**Figure supplement 1—source data 1.** Original and uncropped gels of F$_1$ mice 108–152 for *Figure 4—figure supplement 1A–E*.

**Figure supplement 1—source data 2.** Original and uncropped gels of F$_1$ mice 200–204 for *Figure 4—figure supplement 1A–E*.

**Figure supplement 2.** Genotyping of Obox4 knockout allele in the F$_2$ mice.

**Figure supplement 2—source data 1.** Original and uncropped gels of *Figure 4—figure supplement 2A*.

**Figure supplement 3.** Genotyping of Obox4 and Dux knockout alleles in the F$_2$ mice.

**Figure supplement 3—source data 1.** Original and uncropped gels of F$_2$ mice 187s–201s for *Figure 4—figure supplement 3B*.

**Figure supplement 3—source data 2.** Original and uncropped gels of F$_2$ mice 202s–206s for *Figure 4—figure supplement 3B*.

**Figure supplement 3—source data 3.** Original and uncropped gels of F$_2$ mice 207s–215s for *Figure 4—figure supplement 3B*.

**Figure supplement 3—source data 4.** Original and uncropped gels of F$_2$ mice 216s–240s for *Figure 4—figure supplement 3B*.

**Figure supplement 4.** Genotyping of Obox4 and Dux knockout alleles in the embryos.

**Figure supplement 4—source data 1.** Original and uncropped gels for *Figure 4—figure supplement 4B*.

**Figure supplement 4—source data 2.** Original and uncropped gels for *Figure 4—figure supplement 4E*.

**Figure supplement 5.** *Obox4*/*Dux* double knockdown embryos showed developmental defects.

which showed dysregulation of 2C-genes and TEs targeted by OBOX4 (*Figure 4J*; *Supplementary file 7*). The impaired development and transcriptome of *Dux*<sup>KO</sup>/*Obox4*<sup>KO</sup> embryos at the two-cell stage showed that ZGA was defective in these embryos.

As large numbers of *Obox4*<sup>KO</sup>/*Dux*<sup>KO</sup> embryos are inaccessible due to technical limitations, we adopted an alternative approach to capture high-quality transcriptomes upon *Dux*/*Obox4* depletion. Microinjection of antisense oligonucleotide (ASO) into male pronuclei of zygotes was performed to knockdown *Obox4* and *Dux* (*Figure 4K*, *Figure 4—figure supplement 5A–C*). Monitoring development until 4.5 dpc showed that *Obox4* single knockdown resulted in moderate developmental defects, with almost 50% of embryos reaching the blastocyst stage. Similarly, blastocyst formation was preserved in nearly 25% of *Dux* single knockdown embryos, which is consistent with previously reported *Dux* knockdown/knockout experiments (*De Iaco et al., 2017*; *Chen and Zhang, 2019*; *Guo et al., 2019*; *De Iaco et al., 2020*; *Bosnakovski et al., 2021*). The *Obox4*/*Dux* double knockdown (DKD) markedly compromised blastocyst formation, resulting in more than 80% of embryos degenerated or arrested before the 4C stage, and less than 20% manifested morula-like morphology

(*Figure 4K and L*). The DKD embryos with morula-like morphology manifest a dysregulated transcriptome characterized by a failure of expressing morula specific genes and activation of apoptosis related pathways, suggesting that the development was dysfunctional instead of delayed (*Figure 4— figure supplement 5D–G*).

The consistency among constitutive knockout in SCNT embryos, living mice, and ASO-mediated transient depletion demonstrated that the expression of *Obox4* and *Dux* is collectively important for preimplantation development, and that *Obox4* is capable of promoting ZGA in a *Dux*-independent manner. Collectively, these data show that *Obox4* promotes mouse preimplantation development in the absence of *Dux*.

## OBOX4 promotes 2C-gene expression upon depletion of DUX

As *Obox4* has been shown to activate 2C-genes and TEs in mESCs, and embryonic depletion of OBOX4 impairs 2C-genes and TEs activation, we asked whether the impairment can be ameliorated by restoration of OBOX4. First, we determined whether the presence of OBOX4 can rescue the developmental arrest of *Obox4*/*Dux* DKD embryos. Codon-optimized *Obox4* mRNA without the ASO target motif was produced using in vitro transcription. The rescue experiment was performed by means of co-microinjection of *Obox4* mRNA with *Obox4*/*Dux* DKD ASOs (*Figure 5A*). Restoration of OBOX4 in *Obox4* mRNA-microinjected embryos was confirmed using immunofluorescence staining at the 2C stage (*Figure 5B*). As expected, development monitoring showed that *Obox4*/*Dux* DKD embryos failed blastocyst formation at 4.5 dpc, whereas among *Obox4* mRNA-rescued embryos, blastocyst formation was retained at a similar level to that observed in the *Dux* single knockdown experiment (*Figures 4K, L and 5C, D*). RNA-seq of 2C-embryos revealed exacerbated dysregulation of 2C-genes and TEs by *Obox4*/*Dux* DKD compared to single knockdowns, and the dysregulated transcriptome of DKD was rescued by resupplying OBOX4, revealed by the number of up- and down-regulated genes (*Figure 5E and F*). Differential expression analysis revealed that 58% (102/175) of the downregulated 2C-genes in DKD embryos were rescued, including 2C stage markers and important ZGA factors (*Figure 5G and H*). These results showed that OBOX4 redundantly activates 2C genes to promote ZGA under DUX depletion.

## *Obox4*/*Dux* redundancy is distinct among the *Obox* family

Recently, *Ji et al., 2023* demonstrated that OBOX family proteins are important regulators of ZGA in a DUX-independent manner . Interestingly, they showed that maternal (*Obox1/2/5/7*) and zygotic (*Obox3/4*) *Obox* members seem to be redundant as loss of either is compatible with embryogenesis. This prompted us to ask whether *Obox4* targets are distinctive from that of other *Obox* family members. Comparative analysis revealed distinctive patterns of *Dux*/*Obox3*/*Obox4*/*Obox5*-induced transcriptomes in mESCs (*Figure 6A*). While 2C-genes induced by *Obox3* and *Obox5* overlap substantially, the majority of them were not induced by *Obox4* (*Figure 6B*). The expression changes of *Obox3*/*Obox5* induced 2C-genes are highly correlated, characterized by coefficient of determination $R^2$ (*Figure 6C*), whereas the *Obox4*-induced transcriptome poorly correlated with those of *Obox3* and *Obox5* (*Figure 6D and E*; *Supplementary file 8*). Except for MERVL, TEs induced by *Obox3* and *Obox5* were not induced by *Obox4* (*Figure 6D and E*; *Supplementary file 8*). Similarly, transcriptomes are poorly correlated between *Dux*/*Obox4* DKD and *Obox* maternal/zygotic knockout (mzKO) 2C embryos, with downregulation of distinct groups of 2C-genes (*Figure 6F and G*). The transcriptome of *Obox4* knockdown embryos showed a higher correlation with *Dux*/*Obox4* DKD embryos than *Obox* mzKO embryos (*Figure 6G and H*; *Supplementary file 9*), suggesting that the functional redundancy to *Dux* is a distinct feature of *Obox4* among the *Obox* family.

## Discussion

Starting with a transcriptionally inert genome, the eutherian ZGA engages in massive yet coordinated expression of genes driven by TE LTRs. In mice, it has been reported that the transcription factor DUX accesses and opens condensed chromatin of the MERVL LTR loci, which results in the activation of downstream ZGA genes (*De Iaco et al., 2017*; *Hendrickson et al., 2017*; *Whiddon et al., 2017*). However, the establishment of fertile *Dux* knockout mice suggests that ZGA is redundantly driven by other transcription factor(s) (*Chen and Zhang, 2019*; *Guo et al., 2019*; *De Iaco*

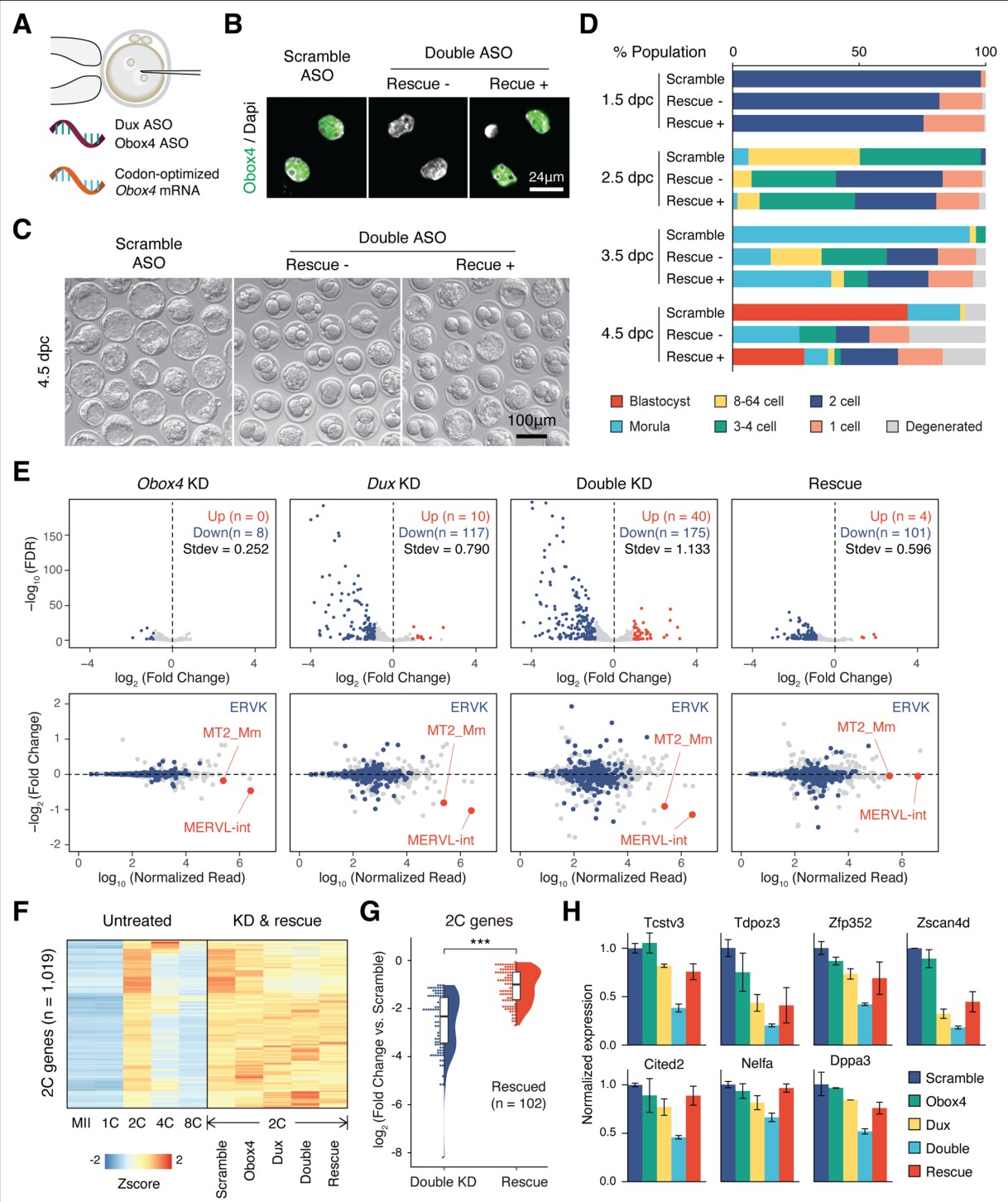

**Figure 5.** OBOX4 promotes zygotic genome activation (ZGA) in the absence of DUX. (**A**) Schematic of the double knockdown rescue experiment. Male pronuclei of zygotes were injected with antisense oligonucleotide (ASO) targeting *Obox4* and *Dux* transcripts as well as in vitro-transcribed codon-optimized *Obox4* mRNA. (**B**) Immunofluorescence staining of OBOX4 in early two-cell embryos microinjected with scrambled ASO, double ASO (*Obox4* and *Dux*), or double ASO with codon-optimized *Obox4* mRNA. (**C**) Representative picture of knockdown and rescue embryos at 4.5 days post coitum (dpc). Codon-optimized *Obox4* mRNA injection rescued blastocyst formation in ASO knockdown embryos. (**D**) The percentages of embryonic stages observed at 1.5 dpc, 2.5 dpc, 3.5 dpc, and 4.5 dpc. The plot represents the sum of three independent experiments, with 80–100 embryos per condition. (**E**) Upper panel: volcano plot showing the results of differentially expressed gene (DEG) analysis of 2C-genes in knockdown and rescue embryos compared to that in scramble ASO-injected embryos. Standard deviations of log$_2$(fold-change) were used to represent the degree of transcriptome dysregulation. Stdev, standard deviation. Lower panel: MA plot of differential expression of transposable elements (TEs) in knockdown and recue

*Figure 5 continued on next page*

*Figure 5 continued*

embryos compared to that in scramble ASO-injected embryos. MERVK elements and MERVL are highlighted. n = 3 biological replicates. (**F**) Heatmaps of the expression of 2C-genes in preimplantation embryos, knockdown 2C-embryos, and rescue 2C-embryos. (**G**) Rain plot displaying the expression change distribution of rescued 2C-genes in double knockdown and recue 2C-embryos. (**H**) Bar plots showing the expression of representative 2C-genes in knockdown and recue embryos; n = 3 biological replicates.

*et al., 2020*; *Bosnakovski et al., 2021*). *Obox* transcripts were first discovered in gonads (*Rajkovic et al., 2002*) and were later found to be highly abundant in mouse 2C-embryos (*Ge, 2017*). Despite being important regulators of ZGA (*Ji et al., 2023*), the biological significance of distinct *Obox* genes remains elusive. The *Obox* family has 67 members clustered in the sub-telomeric region of mouse chromosome 7 and is further divided into six subfamilies: *Obox1*, *Obox2*, *Obox3*, *Obox4*, *Obox5*, and *Obox6*, with *Obox4* constituting nearly 60% (39 of 67) of the family population (*Zhong and Holland, 2011a*; *Wilming et al., 2015*). The role of *Obox4* is particularly mysterious as it remains to be determined whether *Obox4* is a functional protein-coding gene. In the present study, we found that *Obox4* is a bona fide multicopy protein-coding gene that encodes a homeodomain-containing protein that potently induces ZGA gene expression. Using multiple genetic knockout models, we showed that concomitant depletion of *Dux* and *Obox4* was hardly compatible with embryogenesis as *Dux/Obox4* double knockout mESCs subjected to SCNT failed to produce pups, and *Dux/Obox4* double knockout mouse was produced at a sub-Mendelian ratio. Consistently, *Obox4/Dux* DKD markedly compromised preimplantation development, but single knockdown of either gene was tolerated. DKD embryos exhibited severely dysregulated transcriptomes, characterized by MERVL and ZGA gene activation defects. We also characterized the molecular mechanisms underlying the biological significance of *Obox4* as well as its relevance to *Dux*. OBOX4 directly binds to MERVL (the target of DUX) and MERVK loci and activates specific MERVL and MERVK elements in a *Dux*-independent manner. In summary, our results highlight that OBOX4 is a transcription factor that is functionally redundant to DUX during ZGA.

Despite the narrow time window, ZGA takes place as a stepped program. A wave of mild and promiscuous transcription of 'minor ZGA' is commenced first, followed by 'major ZGA' characterized by transcriptional bursting of a more defined set of genes (*Tadros and Lipshitz, 2009*). While major ZGA in large mammals usually launches multiple cell cycles after minor ZGA, it is worth noting that mouse ZGA adopts an unique 'early genome activator' strategy where the timing of minor and major ZGA converges at the early two-cell stage (*Svoboda, 2018*). Overlapping minor and major ZGA right after fertilization allows prompt initiation of the zygotic program and accelerated preimplantation development – a potentially favorable reproductive advantage in rodents. Multiple shared features of *Dux* and *Obox4* imply that these genes have adapted to this particular developmental scheme to serve redundantly as key ZGA factors. Qualitatively, the functional redundancy between DUX and OBOX4 buffers sporadic mutations or defects that could arise during ZGA. Their lack of introns enables the earliest possible translation under ZGA-associated splicing deficiency (*Shen et al., 2021*). Quantitatively, both genes are tandemly arrayed in clusters, which assures rapid accumulation of their transcripts. Taken together, it seems that mouse ZGA may represent an evolutionary innovation where the first embryonic program shows surprising plasticity, likely supported by the expression of multiple *Obox* family members throughout peri-ZGA stages (*Figure 6I*). Indeed, concomitant loss of *Dux* and nearly all *Obox4* copies only partially compromises the ZGA transcriptome, demonstrating that mouse ZGA is so strongly canalized that even such a potent perturbance in genes makes very little difference to the phenotype.

Unlike *Dux*, which is structurally and functionally conserved throughout placentalia (*Leidenroth and Hewitt, 2010*), the ancestral locus of the *Obox* family appears to have undergone mouse-specific duplication and generated a gene cluster that is collectively syntenic to the *Tprx2* locus in other mammals (*Maeso et al., 2016*). Despite the distal homology between *Obox4* and *Tprx2*, it has been recently reported that human *TPRX2* is expressed in eight-cell embryos (*Madissoon et al., 2016*). Interestingly, TPRX2 was shown to cause defective ZGA upon embryonic depletion and bind important ZGA genes in hESCs (*Zou et al., 2022*), suggesting that the *Tprx2* locus might have undergone functionally convergent evolution despite its divergent genetic context. However, due to the distinction of minor and major ZGA timing across species, whether *TPRX2* plays a similar role in humans to *Obox4* in mice regarding the redundancy to *Dux* (*DUX4* in humans) warrants careful examination. Nevertheless,

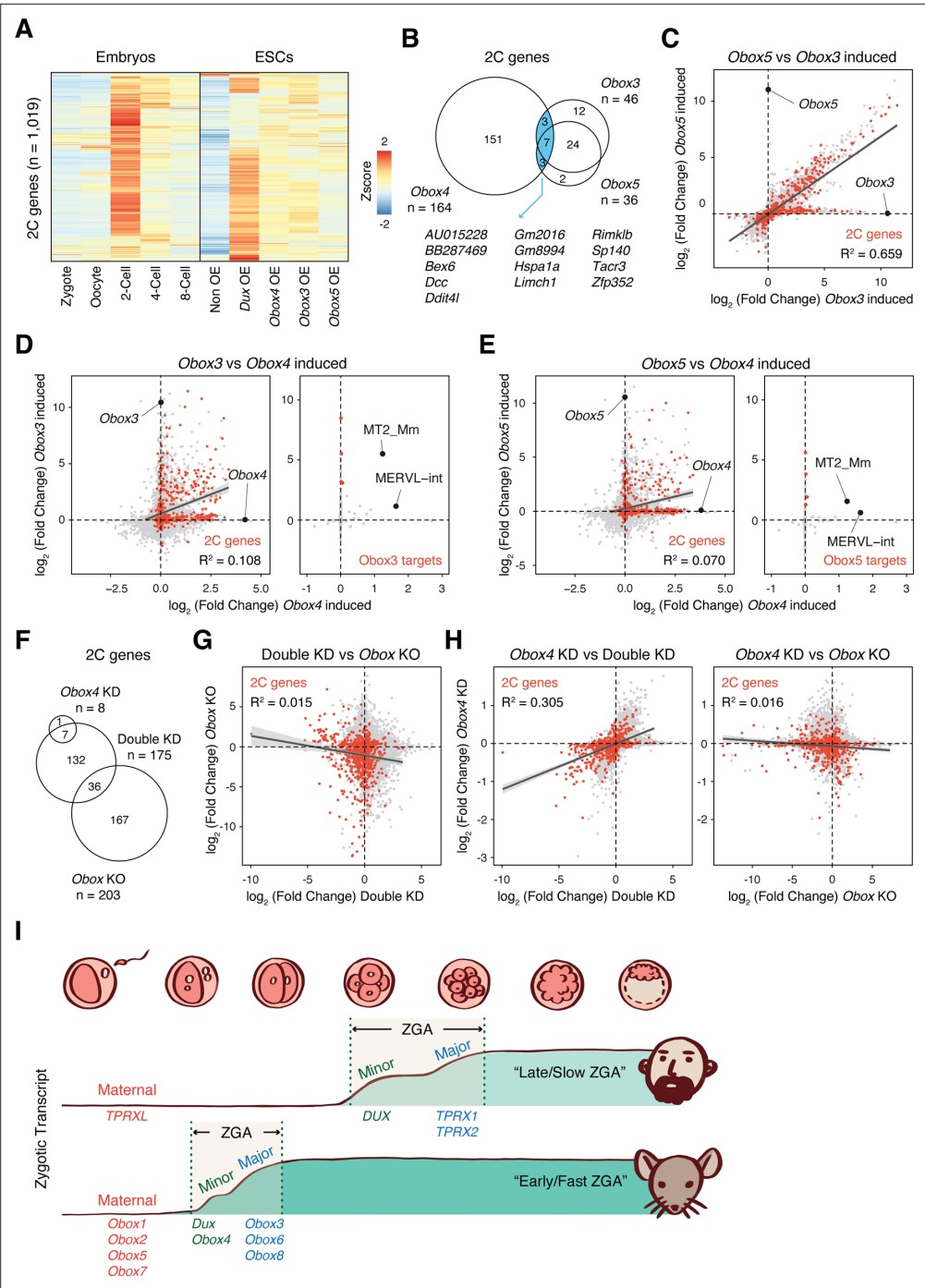

**Figure 6.** OBOX4-mediated DUX redundancy is distinct among the OBOX family. (**A**) Heatmaps of the expression of 2C-genes in preimplantation embryos, non-induced mouse embryonic stem cells (mESCs), and *Dux/Obox4/Obox3/Obox5*-induced mESCs. (**B**) Venn diagram showing overlap of 2C-genes induced by ectopic expression of *Obox3*, *Obox4*, and *Obox5* in mESCs. (**C**) Scatterplot showing per-gene expression changes in Obox3 versus Obox5-induced mESCs. 2C-genes are highlighted in red. (**D**) Scatterplot showing per-gene (left panel) and per-transposable element (right panel) expression changes in *Obox3* and induced versus *Obox4*-induced mESCs. 2C-genes and *Obox3*-induced transposable elements (TEs) are highlighted in red. (**E**) Scatterplot showing per-gene (left panel) and per-transposable element (right panel) expression changes in *Obox5* and induced versus *Obox4*-induced mESCs. 2C-genes and *Obox3*-induced TEs are highlighted in red. (**F**) Venn diagram showing overlap of 2C-genes downregulated upon *Obox4* knockdown, *Dux/Obox4* double knockdown, and *Obox* maternal-zygotic knockout. (**G**) Scatterplot showing per-gene expression changes in *Obox* maternal-

*Figure 6 continued on next page*

*Figure 6 continued*

zygotic knockout versus *Dux/Obox4* double knockdown 2C embryos. 2C-genes are highlighted in red. (**H**) Scatterplot showing per-gene expression changes in *Obox4* knockdown versus *Dux/Obox4* double knockdown (left panel) and *Obox* maternal-zygotic knockout (right panel) 2C embryos. 2C-genes are highlighted in red. (**I**) Schematic model of different zygotic genome activation (ZGA) strategy employed by human and mouse. *Obox* family members with redundant functions are expressed in high dose at peri-ZGA stages to ensure rapid ZGA. *Obox4* evolved as a divergent *Obox* family member that provides functional redundancy to *Dux*.

considering that LTR domestication is functionally convergent in preimplantation development (*Guo et al., 2024*), it might be helpful to speculate that homeobox gene-mediated LTR activation is a common principle of ZGA. Meanwhile, because the repressive epigenetic landscape is barely established during minor ZGA, this may allow transcription factors like DUX and OBOX to potently shape the zygotic transcriptome by activating LTR elements. However, ZGA gene induction and epigenetic reprogramming can only partially recapitulate the 2C transcriptome in mESCs (*Posfai et al., 2021*; *Hu et al., 2023*), suggesting that ZGA is a collective outcome of specific parental, zygotic, and epigenetic factors that may or may not be present in pluripotent stem cells. Indeed, OBOX4 is equally potent as DUX at activating a 2C::tdTomato reporter but less potent at inducing 2C-genes in mESCs, which may suggest that the repressive epigenetic landscape is less permissive to OBOX4 than to DUX. Thus, attempts to model ZGA in vitro need to be carefully formulated to address these fundamental distinctions between embryos and cultured cells.

# Materials and methods

## Key resources

The sequences of primers, ASOs, and sgRNAs used in this study are provided in *Supplementary files 11 and 12*. A complete list of the key reagents and biological samples is provided in *Supplementary file 13*.

## Mice

Male/female BDF1 and female BALB/c mice were purchased from Japan SLC Inc. The mice were fed regular chow and housed in a controlled room under a 14 hr/10 hr light/dark cycle at 22°C. All animal experiments were approved by the Animal Care and Use Committee of Keio University and the Animal Experimentation Committee at the RIKEN Tsukuba Institute and conducted in compliance with the Keio University Code of Research Ethics (License #11045-4) and the RIKEN's guiding principles (T2023-Jitsu015).

## Cell culture

All mouse mESC lines were cultured in Dulbecco's Modified Eagle Medium (DMEM) (Cat# 08488-55, Nacalai Tesque) supplemented with 10% fetal bovine serum (lot# S10581S1820, Biowest), 1× GlutaMAX (Cat# 35050061, Gibco), 1× penicillin-streptomycin (Cat# 15140163, Gibco), 1× non-essential amino acids (Cat# 11140050, Gibco), 1× sodium pyruvate (Cat# S8636, Merck), 50 μM 2-mercaptoethanol (Cat# 21985023, Gibco), 1 μM CHIR99021 (Cat# 034-23103, Wako), 3 μM PD0325901 (Cat# 168-25293, Wako), and mouse leukemia inhibitory factor produced in-house. The culture medium was supplemented with iMatrix-511 silk (recombinant human laminin-511 E8 fragment) (Cat# 892021, Matrixome) at a concentration of 125 ng.cm$^{-2}$ culture vessel area, before seeding mESCs into it. The cells were cultured in a humidified atmosphere containing 5% $CO_2$ at 37°C, with the media changed every 48 hr.

TBLCs were generated by culturing mESCs in an ordinary medium supplemented with 2.5 nM PLaB (Cat# 16538, Cayman Chemicals). Cells were sub-cultured every 2–4 days, at a seeding density of 1 × 10$^5$ cm$^{-2}$ culture vessel area. After five rounds of sub-culture, the cells were deemed TBLCs.

SP2/O-Ag14 myeloma and primary clones of hybridomas were cultured in GIT medium (Cat# 63725715, Wako) supplemented with recombinant human interleukin-6 (IL-6) (Cat# 20006, Pepro-Tech). Cells were cultured in a humidified atmosphere containing 5% $CO_2$ at 37°C and sub-cultured every day at a seeding density of 3×10$^5$ mL$^{-1}$. For monoclonal antibody production, hybridomas were

cultured in a hybridoma serum-free medium supplemented with IL-6. The cells were cultured in a humidified atmosphere containing 5% $CO_2$ at 37°C until they were over-confluent.

## Cell transfection

Plasmid transfections were performed using the jetOPTIMUS DNA Transfection Reagent (Cat# 101000025, Polyplus), according to the manufacturer's instructions. Briefly, mESCs were seeded at a cell density of $8 \times 10^4$ cm$^{-2}$ culture vessel area. After 20 min, the jetOPTIMUS reagent and plasmids were diluted in jetOPTIMUS buffer, incubated for 10 min at room temperature, and then applied to the cell culture. After 48 hr, the cells were collected for downstream experiments.

## cDNA synthesis and cloning

Total RNA was extracted from TBLCs using ISOGEN (Cat# 311-02501, Nippon Gene), according to the manufacturer's instructions. Briefly, 500 µL ISOGEN was added to $1.5 \times 10^6$ freshly harvested TBLCs. After 5 min of incubation at room temperature, 100 µL of chloroform (Cat# 03802606, Wako) was added to the cells and mixed by means of vigorous shaking. After 2 min of incubation at room temperature, the samples were centrifuged at $12,000 \times g$ for 15 min at 4°C. The upper aqueous phase was then transferred to a new tube and mixed with 240 µL isopropanol (Cat# 15-2320, Merck) to precipitate the RNA. After 5 min of incubation at room temperature, the samples were centrifuged at $12,000 \times g$ for 15 min at 4°C. The supernatant was then discarded, following which the RNA pellets were washed with 70% ethanol (Cat# 057-00456, Wako). The RNA pellets were then air-dried and dissolved in RNase-free water. The RNA solution was subjected to DNase treatment using TURBO DNA-*free* kit (Cat# AM1907, Thermo Fisher), according to the manufacturer's instructions, to remove genomic DNA carryover. RNA (1 µg) was reverse transcribed to cDNA using the Transcriptor First Strand cDNA Synthesis Kit (Cat# 04379012001, Roche), according to the manufacturer's instructions. PrimeSTAR Max DNA Polymerase (Cat# R045A, TaKaRa) and ProFlex PCR System (Cat# 4484073, Thermo Fisher) were used to amplify the sequence of interest from the cDNA. NEBuilder HiFi DNA Assembly Kit (Cat# E2621L, NEB) was used to clone the sequence of interest into plasmid backbone.

## Immunofluorescence

For embryos, middle-2C-embryos were collected at 1.5 dpc. The zona pellucida was removed by treating the embryos with acidic Tyrode's solution (Cat# MR-0040-D, Merck). The embryos were fixed in 4% paraformaldehyde (Cat# 09154-14, Nacalai Tesque) in phosphate-buffered saline (PBS) for 10 min. Following three washes in PBS, fixed embryos were permeabilized with 0.1% Triton X-100 (Cat# 1610407, Bio-Rad) in PBS for 20 min at room temperature. Following three washes in PBS, permeabilized embryos were blocked in 2% bovine serum albumin (BSA; Cat# 011-27055, Wako) in PBS for 20 min at room temperature. The blocked embryos were incubated with primary antibodies diluted in 2% BSA in PBS at 4°C overnight. Following three washes in PBS, the embryos were incubated with 2% BSA in PBS containing 1:200 diluted 4',6-diamidino-2-phenylindole (DAPI) (Cat# 19178-91, Nacalai Tesque) and 1:500 diluted Alexa Fluor-conjugated anti-mouse IgG1 secondary antibody (Cat# A-21127, Thermo Fisher) or Alexa Fluor-conjugated anti-mouse IgG2a secondary antibody (Cat# A-21131, Thermo Fisher) for 1 hr at room temperature. Following three washes in PBS, the embryos were transferred into liquid paraffin-covered PBS drops on a glass-bottom dish (Cat# D11130H, Matsunami).

For culturing, cells seeded on glass-bottom chamber slides (Cat# SCS-N02, Matsunami) were fixed with 4% paraformaldehyde in PBS for 10 min. After three washes in PBS, the fixed cells were permeabilized with 0.1% Triton X-100 in PBS for 15 min at room temperature. Following three washes in PBS, permeabilized cells were incubated with primary antibodies diluted with 2% skim milk in PBS for 30 min at room temperature. Following three washes in PBS, the cells were incubated with 2% BSA in PBS containing 1:500 diluted Alexa Fluor-conjugated secondary antibody and 1:200 diluted DAPI for 1 hr at room temperature. Following three washes in PBS, the chamber was removed and the slides were mounted with ProLong Glass Antifade Mountant (Cat# P36982, Thermo Fisher). Fluorescence images were taken using a confocal laser scanning microscope (Cat# FV3000, Olympus).

## Western blot

Freshly harvested cells were resuspended in PBS (Cat# 14249-95, Nacalai Tesque) and lysed by means of sonication. Whole-cell lysates were mixed with sample buffer containing reducing reagent (Cat# 09499-14, Nacalai Tesque) and boiled at 95°C for 5 min. Protein samples were loaded on a 10% tris-glycine gel, run in AllView PAGE Buffer (Cat# DS520, BioDynamics Laboratory), and then transferred to nitrocellulose membranes (Cat# 10600003, Cytiva) using a Power Blotter System (Cat# PB0012, Thermo Fisher). The membranes were blocked with PBST containing 2% skim milk (Cat# 4902720131292, Morinaga) at room temperature for 15 min. The membranes were incubated with primary antibodies in PBST containing 2% skim milk at room temperature for 30 min. After three washes with PBST, the membranes were incubated with secondary antibodies in PBST containing 2% skim milk at room temperature for 15 min with shaking. After three washes with PBST, the membranes were incubated with ECL reagents (Cat# RPN2232, Cytiva) and exposed to an X-ray film (Cat# 28906839, Cytiva) or detected using a digital chemiluminescence imager (Cat# 17001402JA, Bio-Rad).

## qPCR

qPCR was performed using TB Green Fast qPCR Mix (Cat# RR430A, TaKaRa) according to the manufacturer's instructions. The qPCR reactions were carried out and the signals were detected using a real-time PCR system (Cat# TP950, TaKaRa). For RT-qPCR, first-strand cDNA of mESCs total RNA was produced as described in the section 'cDNA synthesis and cloning'. Primers targeting MERVL Gag consensus sequence were used to detect MERVL transcript. Primers targeting β-actin mRNA were used as internal control in all qPCR assays performed in this study.

## smFISH

smFISH probes targeting MERVL were designed and synthesized by LGC Biosearch Technologies as previously described (*Sakashita et al., 2023*). smFISH followed by immunofluorescence staining was performed according to the manufacturer's instructions. Briefly, Quasar 570-labeled probes were hybridized against MERVL RNA at 37°C for 16 hr, followed by immunofluorescence staining described in the section 'Immunofluorescence', without blocking step to prevent RNase contamination. The slides were mounted with ProLong Glass Antifade Mountant (Cat# P36982, Thermo Fisher). Fluorescence images were taken using a confocal laser scanning microscope (Cat# FV3000, Olympus).

## Flow cytometry

The cells were digested with 0.25% trypsin-EDTA (Cat# 25200072, Gibco) at 37°C for 5 min, and then resuspended in FluoroBrite DMEM (Cat# A1896701, Gibco). The cell suspension was filtered through a 35 µM cell strainer (Cat# 352235, Falcon). The samples were analyzed using a cell sorter (Cat# SH800Z, Sony). Data analysis was performed using the Sony SH800Z cell sorter software.

## Generation of transgenic mESC lines

The 2C::tdTomato reporter cell line was generated by transfection of EB3 mESCs (Cat# AES0139, RIKEN BRC Cell Bank) with a linearized 2C::tdTomato reporter plasmid (Cat# 40281, Addgene). After 48 hr, the cells were subjected to 500 µg.mL$^{-1}$ hygromycin (Cat# 08906151, Wako) for 7 days. The selected cells were then seeded at a density of $2 \times 10^2$ cm$^{-2}$ in culture medium containing 250 µg.mL$^{-1}$ hygromycin. Single-cell clones were picked and expanded after 7 days.

A tetracycline-controlled transcription activation system and a piggyBac transposon system were used as previously described (*Takeuchi et al., 2022*) to generate an *Obox4*-inducible cell line. Briefly, pPB-TRE-3×FLAG-Obox4, pPB-CAG-rtTA3G-IRES-Hygro, and pCMV-HyPBase-PGK-Puro plasmids were co-transfected into EB3 mESCs. After 48 hr, the cells were subjected to selection with 500 µg.mL$^{-1}$ hygromycin and 500 µg.mL$^{-1}$ G418 (Cat# 07805961, Wako) for 7 days. The selected cells were then seeded at a density of $2 \times 10^2$ cm$^{-2}$ in a culture medium containing 250 µg.mL$^{-1}$ hygromycin and 250 µg.mL$^{-1}$ G418. Single-cell clones were picked and expanded after 7 days.

## Generation of monoclonal antibodies

The anti-OBOX4, anti-DUX, and anti-GAG monoclonal antibodies were produced as previously described (*Guo et al., 2021*; *Supplementary file 11a*). Briefly, BALB/c mice were immunized by means of intraperitoneal injection of glutathione S-transferase-fused protein-of-interest (PoI). The mice were

routinely immunized until their sera tested positive for the PoI-fused maltose binding protein (MPB), after which two boosting immunizations were performed. Following boosting, splenocytes from immunized mice were fused with SP2/O myeloma and cultured in hypoxanthine-aminopterin-thymidine medium for 10 days to select for hybridomas. Hybridomas were subsequently screened using ELISA against PoI-fused MBP. The positive polyclonal hybridomas were monoclonalized and expanded. For anti-OBOX4 hybridomas, additional screening using OBOX2-fused MBP was conducted. The sequence of OBOX4 was significantly divergent from those of the other OBOX members, and the monoclonal antibodies were negative for cross-reactivity with OBOX2 (*Supplementary file 11b and c*). The N-terminal 100 amino acids of OBOX4 and OBOX2 were used for immunization and cross-reactivity examination, respectively (*Supplementary file 11d*). The culture supernatants of monoclonal hybridomas were sterilized by passing them through a 0.22 µM pore size filter (Cat# 430769, Corning) and used directly as an antibody solution in other assays.

## Generation of knockout mouse lines

Single-guide RNAs (sgRNAs) designed on flanking and inside of the *Obox4* cluster were synthesized using a Precision gRNA Synthesis Kit (Cat# A29377, Thermo Fisher) according to the manufacturer's instructions. 150 ng.mL$^{-1}$ sgRNA and 125 ng.µL$^{-1}$ Alt-R S.p. Cas9 Nuclease (Cat# 1081058, Integrated DNA Technologies) diluted with HEPES-buffered KSOM were electroporated into BDF1×B6 in vitro fertilized embryos with an electroporator (Cat# NEPA21, NEPA GENE). Treated embryos were cultured in KSOM for 18 hr, and embryos that reached the two-cell stage were transferred into the oviducts of pseudopregnant ICR female mice on day 0.5. Pups were retrieved on day 19.5 and genotyped with specific PCR primers.

## Generation of knockout mESC lines

Plasmids encoding Cas9 and sgRNA flanking and inside of the *Obox4* cluster were constructed to perform knockout experiment. The *Obox4* knockout cell line was generated by means of co-transfection of EB3 mESCs with the pX330-Puro-Obox4KO-3, pX330-Puro-Obox4KO-4, pX330-Puro-Obox4KO-6, and pX330-Puro-Obox4KO-7 plasmids. After 18 hr, the cells were subjected to selection with 1 µg.mL$^{-1}$ puromycin (Cat# 16023151, Wako) for 54 hr. The selected cells were then seeded at a very low density in culture medium without antibiotics. Single-cell clones were picked and expanded after 7 days and then subjected to genotyping.

Following a previously published protocol (*Chen and Zhang, 2019*), plasmids encoding Cas9 and sgRNAs flanking the *Dux* cluster were constructed to perform knockout experiment. The *Dux* knockout cell line and the *Obox4/Dux* double knockout cell line were generated by means of co-transfection of EB3 mESCs and *Obox4* knockout cell lines with pX330-DuxKO-A and pX330-DuxKO-B plasmids. After 48 hr, the cells were seeded at a very low density. Single-cell clones were picked and expanded after 7 days and then subjected to genotyping.

While monoallelic *Obox4* deletion was detected in 49.1% (335/682) of the CRISPR-Cas9-edited clones, biallelic deletion was only detected in 0.44% (3/682) of the population, suggesting that removal of *Obox4* allele was associated with severe genetic toxicity. In contrast, 36.7% (47/128) monoallelic and 4.7% (6/128) biallelic deletions were detected in *Dux* knockout clones.

## Genotyping and copy number examination

For mouse and culture cell, mouse right hindlimb toe or tail tip, or 1 × 10$^6$ culture cells were incubated in 400 µL ProK solution containing 20 mM Tris–HCl pH 7.9, 1 mM EDTA, 1% w/v sodium dodecyl sulfate (SDS), 150 mM NaCl, 20 mM trisodium citrate, and 80 µg recombinant proteinase K (Cat# 161-28701, Wako) at 55°C for 2 hr with vigorous shaking. The solution was extracted with 2 volumes of phenol/chloroform/isoamyl alcohol (25:24:1) (Cat# 311-90151, Nippon Gene) twice. DNA in the aqueous phase was precipitated by adding 0.1 volume of 3 M sodium acetate and 2 volumes of 99.5% ethanol (Cat# 057-00456, Wako), snap freeze in liquid nitrogen, and centrifuged at 15,000 × g for 12 min at 4°C. After discarding supernatant, the DNA pellets were washed by 200 µL 70% ethanol, air dried at room temperature, and dissolved in 220 µL TE solution (Tris–HCl pH 7.9, 1 mM EDTA) supplemented with 1 µg RNase A (Cat# 131-01461, Nippon Gene). The solution was incubated at 37°C for 1 hr, then phenol/chloroform/isoamyl alcohol extracted and ethanol precipitated again as described above, to obtain highly purified genomic DNA. The DNA was dissolved in 200 µL TE solution.

For single blastomeres and abnormal embryos, whole-genome amplifications were performed on individual embryos using REPLI-g Advanced DNA Single Cell Kit (Cat# 150363, QIAGEN) according to the manufacturer's instructions. For blastocysts, crude genomic DNA from single blastocysts was prepared according to the method described by *Sakurai et al., 2014* with some modifications. Briefly, single blastocysts in 0.5 µL KSOM medium were transferred to the bottom of 0.1 mL PCR tubes, followed by the addition of 10 µL blastocyst lysis buffer containing 120 µg.mL$^{-1}$ recombinant proteinase K, 100 mM Tris–HCl pH 7.9, 100 mM KCl, 0.45% NP-40, and 10 µg.mL$^{-1}$ yeast tRNA (Cat# AM7119, Thermo Fisher). After brief vortex and pulse spin, the tubes were incubated at 55°C for 10 min followed by 95°C for 10 min.

For PCR-based genotyping, 10 ng purified genomic DNA, 10 ng whole-genome amplified DNA, or 2 µL of blastocyst lysate were used as template and amplified with target-specific primers by KOD One PCR Master Mix (Cat# KMM-101, TOYOBO) according to the manufacturer's instructions. Primers targeting mouse *Tardbp* were used as internal control when whole-genome amplification product or blastocyst lysate were used as multiplex PCR template. For copy number examination, 5 ng genomic DNA template was amplified as described in the 'qPCR' section.

## In vitro transcription

The *Obox4* coding sequence was codon-optimized using GeneArt Instant Designer (Thermo Fisher) to remove the ASO target motif and improve translation efficiency. Codon-optimized DNA was synthesized using Prime Gene Synthesis Services (Thermo Fisher). Codon-optimized *Obox4* mRNA was transcribed in vitro using the mMESSAGE mMACHINE T7 Transcription Kit (Cat# AM1344, Thermo Fisher) and then polyadenylated using a poly(A) tailing kit (Cat# AM1350, Thermo Fisher) according to the manufacturer's instructions. The polyadenylated mRNA was purified using the RNeasy Mini Kit (Cat# 74004, QIAGEN), according to the manufacturer's instructions, and dissolved in RNase-free water at a concentration of 100 ng.µL$^{-1}$.

## Pronuclear injection of mouse embryos

Eight-week-old BDF1 female mice were injected with 150 µL of CARD HyperOva (Cat# KYD-010-EX, Kyudo Co, Ltd). After 48 hr, the female mice were injected with 5 IU human chorionic gonadotropin (hCG) (Cat# GONATROPIN, ASKA Animal Health Co Ltd) and mated with BDF1 male mice. After 22 hr, the PN3 zygotes were collected. Cumulus cells were removed by briefly culturing the zygotes in potassium simplex optimization medium (KSOM) (Cat# MR-101-D, Merck) supplemented with 0.3 µg.µL$^{-1}$ hyaluronidase (Cat# H4272, Merck). Embryos were cultured in KSOM medium drops covered with liquid paraffin (Cat# 26137-85, Nacalai Tesque) in a humidified atmosphere with 5% $CO_2$ at 37°C. After 4 hr, PBS containing either 40 µM scramble ASO, 20 µM anti-*Dux* ASO, 20 µM anti-*Obox4* ASO, 40 µM equimolar anti-*Dux*/anti-*Obox4* ASO mixture, or 40 µM equimolar anti-*Dux*/anti-*Obox4* ASO mixture with 100 ng.µL$^{-1}$ codon-optimized *Obox4* mRNA was microinjected into the male pronuclei of zygotes using a microinjector (Cat# 5252000013, Eppendorf). For developmental monitoring, embryos were cultured for 4 days after microinjection and assessed for developmental stage at 18 hr, 42 hr, 66 hr, and 90 hr after microinjection, which corresponded to 1.5 dpc, 2.5 dpc, 3.5 dpc, and 4.5 dpc, respectively.

## Somatic cell nuclear transfer

SCNT was performed as described previously (*Inoue et al., 2020*) using wild-type and knockout mESC lines as nuclear donor cells. They were cultured at a high density until they reached confluency for about 4–6 days before SCNT. BDF1 female mice were superovulated by the injection of 7.5 IU of equine chorionic gonadotropin (eCG, ZENOAQ) and 7.5 IU of hCG (ASKA Pharmaceutical Co, Ltd) at a 48 hr interval. At 15 hr after hCG injection, cumulus–oocyte complexes were collected from the oviducts. After removal of cumulus cells by 0.1% bovine testicular hyaluronidase (Cat# 385931, Calbiochem), oocytes were enucleated in HEPES-buffered KSOM containing 7.5 µg.mL$^{-1}$ cytochalasin B. After culture in KSOM for at least 1 hr, the enucleated oocytes were injected with donor mESCs using a Piezo-driven micromanipulator (Cat# PMM-150FU, Primetech). After culture in KSOM for about 1 hr, the injected oocytes were cultured in $Ca^{2+}$-free KSOM containing 2.5 mM $SrCl_2$, 5 µM latrunculin A (LatA) (Cat# L5163, Merck) with 50 nM trichostatin A (TSA) (Cat# T8552, Merck) for 1 hr. Then, they were cultured in KSOM containing 5 µM LatA and 50 nM TSA for 7 hr. After washing,

the SCNT embryos were cultured in KSOM under 5% $CO_2$ in air at 37°C for 96 hr. The embryos that reached two cells at 24 hr after activation were considered successfully cloned with cell cycle-matched mESCs (G0/G1 phase).

## scRNA-seq analysis

Raw scRNA-seq data was obtained from the dataset of Deng et al. (GSE45719). Quality control and adapter trimming were done using fastp (*Chen et al., 2018*) v0.23.2. Quality-controlled reads were aligned to the GRCm38.p6 reference genome using STAR (*Dobin et al., 2013*) v2.7.9a, with default arguments. Reads were counted against GRCm38.p6 comprehensive gene annotation (*Frankish et al., 2021*) and mm10 repeats from the University of California, Santa Cruz (UCSC) RepeatMasker using Subread (*Liao et al., 2014*) v2.0.1 featureCounts function, and multi-mapping reads were discarded for non-TE features and counted fractionally for TEs. Seurat (*Hao et al., 2021*) v4.1.0 was used to process the read counts of scRNA-seq. Cells with >7.5% mitochondrial reads or <14,000 annotated features were discarded. Expression levels were log-normalized. The expression profiles of homeobox genes listed in HomeoDB2 (*Zhong and Holland, 2011b*) were clustered into 11 *k*-means after *z*-score transformation.

## Ribo-seq analysis

Raw Ribo-seq data was obtained from the dataset of Xiong et al. (GSE165782). Reads were quality-controlled, aligned, and counted as described above. Reads aligned to all protein coding *Obox4* loci were added up to represent translation level of OBOX4.

## Bulk RNA-seq

For embryos, 30 middle-2C-embryos were collected 20 hr after microinjection in each independent biological replicate. The zona pellucida was removed by treating embryos with acidic Tyrode's solution. Libraries were constructed using the SMART-Seq Stranded Kit (Cat# 634442, TaKaRa) and indexed using the SMARTer RNA Unique Dual Index Kit (Cat# 634451, TaKaRa) according to the manufacturer's instructions.

For cell culture, total RNA was prepared using the RNeasy Kit according to the manufacturer's instructions. The RNA solution was subjected to DNase treatment to remove genomic DNA carryover. Using 1 µg total RNA per sample, libraries were constructed with the NEBNext Ultra II Directional RNA Library Prep Kit for Illumina (Cat# E7760L, NEB) and indexed with NEBNext Multiplex Oligos for Illumina (Cat# E6440S, NEB) according to the manufacturer's instructions.

The libraries were quantified with a 2100 Bioanalyzer (Cat# G2939BA, Agilent) using a High Sensitivity DNA Kit (Cat# 5067-4626, Agilent). Quantified libraries were pooled and sequenced using an Illumina NovaSeq 6000 System in 150 bp paired-end mode (Illumina). Base calling and de-multiplexing were performed using the bcl2fastq2 (Illumina) v2.20. De-multiplexed reads were quality-controlled, aligned, and counted as described above. DESeq2 (*Love et al., 2014*) v1.32.0 was used to perform differential expression analysis. Raw RNA-seq reads for TBLCs were downloaded from the dataset of Shen et al. (GSE168728). Raw RNA-seq reads of *Dux* knockout embryos were downloaded from the datasets of Chen and Zhang and De Iaco et al. (GSE121746 and GSE141321, respectively). Raw RNA-seq reads of preimplantation embryos were downloaded from the datasets of Wu et al. (GSE66390). Raw RNA-seq reads of 2C-like mESCs were downloaded from the datasets of Zhu et al. (GSE159623). The published data were analyzed using the same method.

## CUT&RUN-seq

Cultured cells (1 × 10⁵) freshly prepared by trypsin-EDTA digestion were used to perform CUT&RUN with the CUT&RUN Assay Kit (Cat# 86652, Cell Signaling Technology) according to the manufacturer's instructions. Enriched DNA was subjected to library construction using the NEBNext Ultra II DNA Library Prep Kit for Illumina (Cat# E7645L, NEB), with the adapter ligation step performed at 50°C instead of 65°C, to prevent the denaturation of small DNA inserts. The libraries were indexed using NEBNext Multiplex Oligos for Illumina, according to the manufacturer's instructions.

The libraries were quantified, pooled, sequenced, de-multiplexed, and quality controlled as described above. Processed reads were aligned to the GRCm38.p6 reference genome using Bowtie2 (*Langmead and Salzberg, 2012*) v2.4.1, with default settings. Peaks were called in each

biological replicate using the MACS3 (*Zhang et al., 2008*) v3.0.0a6 callpeak function (-f BAMPE). Alignment tracks were first generated using deepTools (*Ramírez et al., 2016*) v3.5.1 bamCoverage function (`--binSize` 10 `--normalizeUsing` CPM `--smoothLength` 30), and then normalized by subtracting the signal from non-immune IgG and wild-type mESCs using the deepTools bamCompare function (`--scaleFactorsMethod` None `--operation` subtract `--binSize` 10 `--smoothLength` 30). ChIPseeker (*Yu et al., 2015*) v1.28.3 was used to annotate the peaks. The MEME suite (*Bailey et al., 2009*) v5.4.1 was used to identify the binding motifs. Heatmaps were generated using deepTools computeMatrix and plotHeatmap functions. Visualization of genomic tracks was performed using trackplot (*Pohl and Beato, 2014*) v1.3.10. Raw ChIP-seq reads of DUXs in mESCs were downloaded from Hendrickson et al. (GSE85632). The published data were analyzed using the same method.

## Quantification and statistical analysis

Descriptive and comparative statistics were employed in the article as described in the figure legends, with the number of replicates indicated. For single-hypothesis testing, significance is defined as a $p$-value$<0.05$ indicated with asterisk (*$p$-value$<0.05$, **$p$-value$<0.01$, ***$p$-value$<0.001$). Error bar represents the standard deviation (SD) of the mean of the replicates. Significant change in gene expression is denoted by greater than onefold difference and less than 0.01 false discovery rate.

## Acknowledgements

We thank all the members of the Siomi Laboratory for their discussions and comments on this work. We thank Takehiko Yokomizo (Department of Biochemistry, Juntendo University) for providing the anti-DYKDDDDK monoclonal antibody (clone 2H8). We also thank Daisuke Motooka (Research Institute for Microbial Diseases, Osaka University) for generating the sequencing data. We are grateful to Azusa Inoue (Center for Integrative Medical Sciences, RIKEN), Katsuhiko Hayashi (Department of Genome Biology, Osaka University), and Therese Solberg (WPI-Bio2Q, Keio University) for their comments on this article. This work was supported by the MEXT Grant-in-Aid for Scientific Research in Innovative Areas (19H05753 to HS and 19H05758 to AO), AMED project to elucidate and control mechanisms of aging and longevity (1005442 to HS), JSPS Grant-in-Aid for Scientific Research KAKENHI (20K21507 to KM and 22H02534 to KI), Mochida Memorial Foundation Research Grant to KM, Sumitomo Foundation Research Grant to KM, Keio University Doctorate Student Grant-in-Aid Program to YG, and the JST Doctoral Program Student Support Fellowship to YG.

## Additional information

### Funding

| Funder | Grant reference number | Author |
| --- | --- | --- |
| Ministry of Education, Culture, Sports, Science and Technology | Grant-in-Aid for Scientific Research in Innovative Areas 19H05753 | Haruhiko Siomi |
| Japan Agency for Medical Research and Development | Project to Elucidate and Control Mechanisms of Aging and Longevity | Haruhiko Siomi |
| Ministry of Education, Culture, Sports, Science and Technology | Grant-in-Aid for Scientific Research in Innovative Areas 19H05758 | Atsuo Ogura |
| Japan Society for the Promotion of Science | Grant-in-Aid for Scientific Research KAKENHI 20K21507 | Kensaku Murano |
| Japan Society for the Promotion of Science | Grant-in-Aid for Scientific Research KAKENHI 22H02534 | Kimiko Inoue |

| Funder | Grant reference number | Author |
|---|---|---|
| Mochida Memorial Foundation for Medical and Pharmaceutical Research | | Kensaku Murano |
| Sumitomo Foundation | | Kensaku Murano |
| Keio University | Student Grant-in-Aid Program | Youjia Guo |
| Japan Science and Technology Agency | Doctoral Program Student Support Fellowship | Youjia Guo |

The funders had no role in study design, data collection and interpretation, or the decision to submit the work for publication.

## Author contributions

Youjia Guo, Conceptualization, Resources, Data curation, Software, Formal analysis, Funding acquisition, Validation, Investigation, Visualization, Methodology, Writing - original draft, Project administration, Writing - review and editing; Tomohiro Kitano, Conceptualization, Data curation, Software, Formal analysis, Validation, Investigation, Visualization, Methodology, Project administration, Writing - review and editing; Kimiko Inoue, Data curation, Validation, Investigation, Visualization, Methodology, Project administration, Writing - review and editing; Kensaku Murano, Resources, Data curation, Supervision, Investigation, Methodology, Project administration, Writing - review and editing; Michiko Hirose, Ten D Li, Narumi Ogonuki, Shogo Matoba, Masayuki Sato, Investigation; Akihiko Sakashita, Investigation, Writing - review and editing; Hirotsugu Ishizu, Resources; Atsuo Ogura, Resources, Data curation, Supervision, Funding acquisition, Validation, Investigation, Methodology, Project administration, Writing - review and editing; Haruhiko Siomi, Conceptualization, Resources, Supervision, Funding acquisition, Project administration, Writing - review and editing

## Author ORCIDs

Youjia Guo (iD) http://orcid.org/0000-0002-4837-9350
Tomohiro Kitano (iD) http://orcid.org/0000-0002-3433-9163
Masayuki Sato (iD) http://orcid.org/0000-0002-5484-4597
Atsuo Ogura (iD) http://orcid.org/0000-0003-0447-1988
Haruhiko Siomi (iD) http://orcid.org/0000-0001-8690-3822

## Ethics

All animal experiments were approved by the Animal Care and Use Committee of Keio University and the Animal Experimentation Committee at the RIKEN Tsukuba Institute and conducted in compliance with the Keio University Code of Research Ethics and the RIKEN's guiding principles. (License #11045-4) and the RIKEN's guiding principles (T2023-Jitsu015).

## Decision letter and Author response

Decision letter https://doi.org/10.7554/eLife.95856.sa1
Author response https://doi.org/10.7554/eLife.95856.sa2

---

# Additional files

## Supplementary files

• Supplementary file 1. List of Z-score and *k*-means clustering of genes based on preimplantation expression profiling data by *Wu et al., 2016* (GSE66390) and *Deng et al., 2014* (GSE45719).

• Supplementary file 2. Deferential expression analysis results of genes and transposable elements in *Obox4* and *Dux* overexpressing mESCs.

• Supplementary file 3. List OBOX4 and DUX bond genes.

• Supplementary file 4. Total numbers of annotated loci in each transposable element family and numbers of loci bond by OBOX4.

• Supplementary file 5. List of mating pairs and genotyping results of *Dux/Obox4* knockout mice.

• Supplementary file 6. Genotyping and development monitoring results of preimplantation

embryos produced by *Dux*<sup>KO</sup>/*Obox4*<sup>Het</sup> × *Dux*<sup>Het</sup>/*Obox4*<sup>Het</sup> mating pair.

• Supplementary file 7. Deferential expression analysis results of genes and transposable elements in *Dux*<sup>KO</sup>/*Obox4*<sup>Het</sup> and *Dux*<sup>KO</sup>/*Obox4*<sup>KO</sup> versus *Dux*<sup>KO</sup>/*Obox4*<sup>WT</sup> 2C embryos.

• Supplementary file 8. Combined deferential expression analysis results of genes and transposable elements in *Obox3/4/5* overexpressing mESCs.

• Supplementary file 9. Combined deferential expression analysis results of genes and transposable elements in *Dux/Obox4* knockdown and *Obox* family knockout 2C embryos.

• Supplementary file 10. Primer sequences used in this study.

• Supplementary file 11. Antisense oligonucleotides sequences used in this study.

• Supplementary file 12. sgRNA sequences used in this study.

• Supplementary file 13. Key resources table.

• MDAR checklist

## Data availability

The RNA-seq and CUT&RUN-seq data generated in this study have been deposited at NCBI Gene Expression Omnibus (GEO) database under the accession code GSE196671.

The following dataset was generated:

| Author(s) | Year | Dataset title | Dataset URL | Database and Identifier |
|---|---|---|---|---|
| Guo Y, Kitano T, Siomi H | 2023 | Obox4 secures zygotic genome activation upon loss of Dux | https://www.ncbi.nlm.nih.gov/geo/query/acc.cgi?acc=GSE196671 | NCBI Gene Expression Omnibus, GSE196671 |

The following previously published datasets were used:

| Author(s) | Year | Dataset title | Dataset URL | Database and Identifier |
|---|---|---|---|---|
| Deng Q, Ramsköld D, Reinius B, Sandberg R | 2014 | Single-cell RNA-Seq reveals dynamic, random monoallelic gene expression in mammalian cells | https://www.ncbi.nlm.nih.gov/geo/query/acc.cgi?acc=GSE45719 | NCBI Gene Expression Omnibus, GSE45719 |
| Xiong Z, Xu K, Lin Z, Kong F, Wang Q, Quan Y, Li L, Xie W | 2022 | Ultrasensitive Ribo-seq reveals translational landscapes during mammalian oocyte-to-embryo transition and pre-implantation development | https://www.ncbi.nlm.nih.gov/geo/query/acc.cgi?acc=GSE165782 | NCBI Gene Expression Omnibus, GSE165782 |
| Shen H, Yang M, Li S Du P | 2021 | Mouse totipotent stem cells captured and maintained through spliceosomal repression | https://www.ncbi.nlm.nih.gov/geo/query/acc.cgi?acc=GSE168728 | NCBI Gene Expression Omnibus, GSE168728 |
| Chen Z, Zhang Y | 2019 | Loss of DUX causes minor defects in zygotic genome activation and is compatible with mouse development | https://www.ncbi.nlm.nih.gov/geo/query/acc.cgi?acc=GSE121746 | NCBI Gene Expression Omnibus, GSE121746 |
| Grun D, De Iaco A | 2019 | RNAseq of DUX KO mouse 2-cell embryos | https://www.ncbi.nlm.nih.gov/geo/query/acc.cgi?acc=GSE141321 | NCBI Gene Expression Omnibus, GSE141321 |
| Peter GH | 2017 | Conserved roles for murine mDUX and human DUX4 in activating cleavage stage genes and MERVL/HERVL retrotransposons | https://www.ncbi.nlm.nih.gov/geo/query/acc.cgi?acc=GSE85632 | NCBI Gene Expression Omnibus, GSE85632 |

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
