## [Editor Report]

This study presents an important finding that Obox4 and Dux act redundantly in regulating zygotic genome activation in mice. The evidence supporting the claims of the authors is convincing. The work will be of interest to researchers interested in early embryo development and epigenetic reprogramming.

---

## [Decision Letter]

**Decision letter after peer review:**

Thank you for submitting your article "Obox4 promotes zygotic genome activation upon loss of Dux" for consideration by *eLife*. Your article has been reviewed by 2 peer reviewers, and the evaluation has been overseen by a Reviewing Editor and Lori Sussel as the Senior Editor. We invite you to revise the manuscript to address the comments from the reviewers.

Essential revisions (for the authors):

1. As Obox cluster KO (PMID: 37459895) gives sound evidence for the role of Obox family proteins in ZGA, comparisons of Obox4 KO and Obox4/Dux DKO with the previous Obox cluster KO will greatly benefit the community. It is essential to perform a comparative analysis to identify similar and/or differentially affected genes between these two studies, using your data and the publicly available data as indicated by Reviewer 2.

2. It will be good to compare how Obox4 and Dux regulate ZGA and 2C genes using the single and double KO models already in hand as indicated by Reviewer 2.

3. Please discuss the several points outlined by Reviewer 1, which could strengthen the manuscript greatly.

*Reviewer #1 (Recommendations for the authors):*

The work has been carefully done and I do not have any suggestions as it seems that the message authors convey is sufficiently supported by multiple lines of evidence. My comments thus concern primarily the interpretation of data and discussion, which I think could be strengthened if the following points would not be overlooked.

1) I think authors should point out the uniqueness of the murine zygotic genome activation because the "minor ZGA" at the 1-cell stage is apparently largely overlapping with the major ZGA at the 2-cell stage, at least when it comes to genes activated during the major ZGA. At the same time, naïve chromatin matures to the transcriptionally repressive state until the late 2-cell stage, which provides an opportunity for Dux and OBOX4 to activate transcription through LTR enhancers.

2) Because of (1), I believe that lessons about ZGA program when induced (inefficiently) in ESCs and activated (100% efficiently) in zygotes should be carefully formulated when it comes to the activation of this program, which is under physiological conditions triggered via maternally provided TFs, which may or may not be expressed in ESCs.

3) The nature of Dux & Obox4 genes should be brought up as both have features implying adaptations for their role and further supporting that they are indeed the key factors of ZGA. In both cases, there are tandemly arrayed genes = this assures rapid accumulation of their transcripts in the earliest phases of the genome activation. They also do not have coding sequences interrupted in introns. As the minor ZGA has been accompanied with deficient splicing, this feature would enable the earliest possible translation of Dux and Obox4 transcripts – clearly a desirable feature in an "early activator" like mouse.

4) Phylogenetic history of the coding sequences and tandem arrays of Dux and Obox4 as well as of MuERV-L suggests that mice represent one evolutionary solution of a surprisingly plastic activation of the first embryonic expression program. In this sense, it might be helpful to speculate what may be common principles and species-specific adaptations of mammalian ZGA.

*Reviewer #2 (Recommendations for the authors):*

Following are specific comments:

1) There are some inconsistent outcomes among different approaches used to perturb Obox4 and Dux. It appears that ASOs and SCNT cause more severe phenotypes as no embryos form blastocysts, or blastocysts with good morphology (Figure 4k, 4b-c). However, Obox4 and Dux DKO embryos can survive at least to the weaning age, suggesting that Obox4/Dux DKO is not 100% penetrant. Could authors discuss what may cause the inconsistencies? Are there any abnormalities associated with the Obox4/Dux DKO mouse? Did authors observe more DKO mice after manuscript submission? Are the Obox4/Dux DKO mice fertile?

2) I fully agree with the statement that "it is critical to examine the functional requirement of OBOX4 and DUX in genetic knockout models" (line 137). Thus, I recommend authors comparing how Obox4 and Dux regulate ZGA and 2C genes using the single and double KO models they already generated instead of focusing on ASO-mediated KD. The genetic KO models are much more convincing than ASO-mediated KD.

3) The authors should compare how Obox4/Dux DKO vs. Obox cluster KO in regulating 2C repeats and ZGA genes. It seems to me that Obox4/Dux double KD or KO causes less severe ZGA defects than Obox cluster KO. For example, Obox/Dux DKD causes ~175 genes down-regulated at 2-cells stage (Figure 5) (this info is missing for the DKO), however, Obox cluster KO causes downregulation of ~530 major ZGA genes (Ji et al., Figure 3). I highly recommend authors performing a comparative analysis to identify similar and/or differentially affected genes between these two studies.

4) Figure 1b, what about expression levels of other Obox genes in TBLCs vs. ESC?

5) Line 79, Obox4 should be partially affected by Dux KO based on Figure 1c and the legend.

6) Figure S1b, it seems that Dux is not expressed in overexpressed cells. Is this due to that Dux is quickly degraded?

7) Figure 2, How similar or different for Obox4 OE vs. Obox3 and 5 OE as reported by Ji et al? Are MT2C_Mm and MT2B2 also induced upon Obox4 OE?

8) Figure 2h, does Dux also induce other Obox genes in addition to Obox4?

9) Figure S3b. Why much fewer peaks (~10%) of Dux CUT&RUN peaks vs. ChIP-seq peaks? Are the missing peaks false negatives?

10) Obox1/3/5 seem also bind to B1/B2 elements in addition to ERVL. How about Obox4? Since ESCs chromatin structure is very different from 2-cell embryos. The authors may consider profiling Obox4 binding in embryos.

11) Since one Obox4 copy with ORF was not depleted according to the KO design, the authors should perform immunostaining to confirm whether Obox4 protein is gone in Obox4 KO embryos.

12) Figure 4i is confusing, why very few Dux-Het/Obox4KO embryos reach to blastocyst at 4.5 dpc? In theory, Dux-Het/Obox4KO should show no embryonic arrest based on Figure 4e-g.

13) Since Obox4 has a high identity to other Obox proteins (Figure S11), authors should demonstrate the antibody specificity.

14) Line 231-233. "However, whether and how other Obox family members contribute to ZGA remains unclear". This is not true, because Ji et al. has showed that Obox is critical in regulating ZGA.

15) Line 56-57. "Nr5a2 that triggers ZGA by activating short interspersed nuclear element (SINE) B1 family". The authors should be cautious about this statement because it has been shown that Nr5a2 had a minimal role in ZGA by maternal-zygotic KO mouse model (https://doi.org/10.1101/2023.01.16.524255).

16) I feel that only a few representative genotyping gel images are sufficient. The authors do not need to show all the raw genotyping gels. In addition, Figure 5b (A+B) gel is noisy and it's difficult to locate the target bands.

17) Typo in Figure S3a, it should be Dux instead of Obox?

---

## [Author Response]

Essential revisions (for the authors):1. As Obox cluster KO (PMID: 37459895) gives sound evidence for the role of Obox family proteins in ZGA, comparisons of Obox4 KO and Obox4/Dux DKO with the previous Obox cluster KO will greatly benefit the community. It is essential to perform a comparative analysis to identify similar and/or differentially affected genes between these two studies, using your data and the publicly available data as indicated by Reviewer 2.2. It will be good to compare how Obox4 and Dux regulate ZGA and 2C genes using the single and double KO models already in hand as indicated by Reviewer 2.3. Please discuss the several points outlined by Reviewer 1, which could strengthen the manuscript greatly.

We would like to thank the editors and reviewers for their encouraging remarks and comments that we believe have truly helped us to improve our manuscript, particularly the suggestion made by Reviewer #1 to discuss the adaptation of the *Obox4* cluster to the early genome activation model employed by mouse, and the request made by Reviewer #2 to compare regulatory targets of *Obox4* to other *Obox* family members. As double KO mating pairs remain unavailable despite repeated breeding attempts, we performed comparative analysis between genes and transposable elements (TEs) affected by the *Dux*/*Obox4* DKD and *Obox* family KO. The analysis showed that *Obox4* has minimal regulatory overlap with other *Obox* family members, suggesting that the functional redundancy to *Dux* is a distinct feature of *Obox4* within the *Obox* family. We have revised the manuscript accordingly. Please find below our point-to-point response to all the concerns and suggestions made by the reviewers.

Point-by-point response to reviewers’ comments:

Reviewer #1 (Recommendations for the authors):The work has been carefully done and I do not have any suggestions as it seems that the message authors convey is sufficiently supported by multiple lines of evidence. My comments thus concern primarily the interpretation of data and discussion, which I think could be strengthened if the following points would not be overlooked.1) I think authors should point out the uniqueness of the murine zygotic genome activation because the "minor ZGA" at the 1-cell stage is apparently largely overlapping with the major ZGA at the 2-cell stage, at least when it comes to genes activated during the major ZGA. At the same time, naïve chromatin matures to the transcriptionally repressive state until the late 2-cell stage, which provides an opportunity for Dux and OBOX4 to activate transcription through LTR enhancers.

We thank the reviewer for the insightful advice. We have revised the manuscript to discuss about this point extensively in the Discussion section.

2) Because of (1), I believe that lessons about ZGA program when induced (inefficiently) in ESCs and activated (100% efficiently) in zygotes should be carefully formulated when it comes to the activation of this program, which is under physiological conditions triggered via maternally provided TFs, which may or may not be expressed in ESCs.

We fully agree that the distinction between embryonic ZGA and the activation of ZGA genes in ESCs need to be carefully addressed. We have revised the manuscript to reflect this distinction (revised manuscript line #68-70) and emphasized it in the Discussion section.

3) The nature of Dux & Obox4 genes should be brought up as both have features implying adaptations for their role and further supporting that they are indeed the key factors of ZGA. In both cases, there are tandemly arrayed genes = this assures rapid accumulation of their transcripts in the earliest phases of the genome activation. They also do not have coding sequences interrupted in introns. As the minor ZGA has been accompanied with deficient splicing, this feature would enable the earliest possible translation of Dux and Obox4 transcripts – clearly a desirable feature in an "early activator" like mouse.

We thank the reviewer for the excellent discussion. We have incorporated and expanded this idea into the discussion together with the reviewer’s suggestion in comment 4 and appended a schematic illustration in the newly added figure (Figure 6I).

4) Phylogenetic history of the coding sequences and tandem arrays of Dux and Obox4 as well as of MuERV-L suggests that mice represent one evolutionary solution of a surprisingly plastic activation of the first embryonic expression program. In this sense, it might be helpful to speculate what may be common principles and species-specific adaptations of mammalian ZGA.

In accordance with our response to comment 3, we have incorporated this into the discussion.

Reviewer #2 (Recommendations for the authors):Following are specific comments:1) There are some inconsistent outcomes among different approaches used to perturb Obox4 and Dux. It appears that ASOs and SCNT cause more severe phenotypes as no embryos form blastocysts, or blastocysts with good morphology (Figure 4k, 4b-c). However, Obox4 and Dux DKO embryos can survive at least to the weaning age, suggesting that Obox4/Dux DKO is not 100% penetrant. Could authors discuss what may cause the inconsistencies? Are there any abnormalities associated with the Obox4/Dux DKO mouse? Did authors observe more DKO mice after manuscript submission? Are the Obox4/Dux DKO mice fertile?

We thank the reviewer for the comment. There are multiple factors that potentially contribute to the more severe phenotype observed in vitro. Some technical concerns could be intrinsic to the adopted technique (e.g. injection induced damage), the material used (ESCs derived from inbred strain, epigenetic barrier), or environmental. However, with the proper controls and by employing multiple techniques to minimize technical biases, the consistency between different in vitro techniques suggest that the phenotype is truly a result of DUX/OBOX4 depletion. Nevertheless, perhaps one of the biggest factors that is inevitable was the culture condition. Even the best in vitro culture condition only partially recapitulates the in utero environment, which is the most critical external factor for embryogenesis. In fact, it has long been recognized in the reproductive field that in vitro culturing elicits a strong stress response and compromises blastocyst formation (DoI: 10.1530/REP-16-0391). While some adverse effects can be ameliorated (e.g. culture under hypoxia instead of normoxia), faithful recapitulation of the in vivo development is still impossible. Due to limitation in equipment, all embryos were cultured under normoxia in our study, which causes oxidative stress. Considering even healthy embryos are susceptible to oxidative stress (DoI: 10.1016/j.rvsc.2020.07.013), the likelihood of blastocyst formation could be much lower in ASO knockdown and SCNT embryos than the knockouts produced by natural mating.

Regarding the Obox4/Dux DKO mouse, unfortunately, despite repeated attempts, we only obtained one DKO mouse in the early generation. We would love to produce more, if it is possible, DKO mice for phenotypic study but the huge colony size required may saturate our mouse production capacity before a mating pair can be obtained.

2) I fully agree with the statement that "it is critical to examine the functional requirement of OBOX4 and DUX in genetic knockout models" (line 137). Thus, I recommend authors comparing how Obox4 and Dux regulate ZGA and 2C genes using the single and double KO models they already generated instead of focusing on ASO-mediated KD. The genetic KO models are much more convincing than ASO-mediated KD.

We completely agree with the reviewer and wish we could perform in-depth transcriptome analysis using genetic KO models. Because DKO mating pairs are not available, it seems that the practical way left to perform such analysis is to collect individual 2C embryos produced by heterozygotic knockouts, separate the two blastomeres, use one of the blastomeres for genotyping and the other for scRNA-seq so that the genotypes can be associated with the transcriptomes (as illustrated in Figure 4—figure supplement 4D). We managed to profile 31 blastomere pairs and pool their scRNA-seq as pseudo-bulk and found that 79 and 5 2C-genes and are downregulated in DuxKO/Obox4Het and DKO versus DuxKO/Obox4WT blastomeres, respectively. MERVL is only downregulated in DKO blastomeres (Figure 4J, Figure 4—figure supplement 4E, and Figure 4—figure supplement 3).

However, due to the extreme technical difficulty, the quality of such an experiment is sub-optimal and only provides a coarse snapshot of the transcriptome of the DKO based on following metrics:

- First, our RNA-seq quality control showed that inter-sample variance between of DKO is sub-optimal. Normally we deem a coefficient of determination (R^2^) greater than 0.95 as a good indicator of the consistency between biological replicates. The R^2^ of DKO RNA-seq is 0.740 (Author response image 1), while for DKD RNA-seq the number is 0.988 (Author response image 1). Although it is believed that genetic knockout models are theoretically superior at resolving the molecular phenotype, in this experiment the quality control suggested that the DKO RNA-seq is more likely to introduce statistic artifacts than those of the DKD (Author response image 1). Unfortunately, repeating this experiment is unlikely to improve the data, as these issues stem from the method itself, such as the low sample number, handling of live separated blastomeres, and the nature of single-cell RNA sequencing etc.

- Second, despite that the RNA-seq of DUX KO 2C show discrepancies between different samples, our ASO KD RNA-seq is highly consistent with these KO RNA-seq, suggesting that the ASO KD approach can reproduce the molecular phenotype of genetic KO studies (DoI: 10.1038/s41588-019-0418-7 and 10.1038/s41422-019-0238-4, also see Author response image 1).

Based on these observations, we concluded that our ASO mediated KD faithfully recapitulated genetic KO models while providing a higher resolution of the transcriptome for in-depth analysis. This became the rationale of using ASO-mediated KD for the comparative analysis.

**Author response image 1. sa2fig1:** Comparison of transcriptome data in different KO and KD experiments. a) Correlations between biological replicates of *Dux*/*Obox4* DKO pseudo-bulk scRNA-seq (left panel) and ASO mediated Dux/Obox4 DKD RNA-seq (right panel). DKO samples bear higher inter-sample variance than DKD samples. 2C-genes are highlighted, inter-sample transcriptome correlations are measured by coefficient of determination (R^2^). b) Correlations of gene expression changes between *Dux*/*Obox4* DKO and DKD RNA-seq. RNA-seq of DKD provides higher statistic power at discovering down-regulated 2C genes. 2C-genes are highlighted. c) Correlations of 2C-gene expression changes between published *Dux* KO RNA-seq and our ASO-mediated *Dux* KD RNA-seq. Our KD transcriptome shows higher agreement with the two KOs produced by Guo *et al.* and Chen *et al.* than between them each other.

3) The authors should compare how Obox4/Dux DKO vs. Obox cluster KO in regulating 2C repeats and ZGA genes. It seems to me that Obox4/Dux double KD or KO causes less severe ZGA defects than Obox cluster KO. For example, Obox/Dux DKD causes ~175 genes down-regulated at 2-cells stage (Figure 5) (this info is missing for the DKO), however, Obox cluster KO causes downregulation of ~530 major ZGA genes (Ji et al., Figure 3). I highly recommend authors performing a comparative analysis to identify similar and/or differentially affected genes between these two studies.

This is a very important point raised by the reviewer, which is highly relevant to the reviewer’s comment 2. We agree that DKO would be the ideal model for addressing this point, but since DKO mating pairs are currently unavailable, we performed a comparative analysis of our Obox4/Dux DKD and the Obox cluster KO. We found minimal overlap between genes regulated by *Dux*/*Obox4* and other *Obox* family members. This suggests that *Obox4* is a distinct *Obox* family member that promotes ZGA primarily through providing redundancy to *Dux*. Note that the definitions of “ZGA genes” and “2C-genes” are distinct, thus the number of genes belonging to these two categories cannot be compared directly (175 2C-genes vs. 530 major ZGA genes). The concept of ZGA gene denotes a gene that is activated, but does not necessarily only express, during ZGA. Accordingly, in the paper by Ji *et al.*, ZGA genes are designated by having FPKM > 5 and greater than threefold upregulation at 1C or 2C comparing to oocytes (DoI: doi.org/10.1038/s41586-023-06428-3, Methods). On the other hand, to be deemed a 2C-gene, a gene needs to only highly express at the 2C stage. In our study, we do not set expression level and foldchange thresholds when defining 2C-genes. Instead, we use Z-score to examine fluctuation of gene expression and perform k-means cluster analysis to identify genes whose high expressions are exclusive to the 2C stage (Figure 2—figure supplement 1A and Supplementary File 1). We believe examining the effects of stage-specific transcription factors on 2C-genes better captures the dynamic impacts caused by these transcription factors. In our analysis, *Obox* mzKO caused a down-regulation of 203 2C-genes, which is only a slightly greater number than the 175 caused by *Dux*/*Obox4* DKD. However, considering that *Obox* mzKO and *Dux*/*Obox4* DKO both caused 2C arrest, we speculate that *Dux*/*Obox4* and other *Obox* family members collectively provide broader coverage of ZGA gene activation than individually. Losing either *Dux*/*Obox4* and *Obox1/2/3/5/6/7* axis will compromise a significant portion of ZGA program, which cannot be rescued by the other. We have included results of the analysis in a new figure and supplementary tables with detail comparison of genes/TEs affected by *Dux*, *Obox4*, and other *Obox* family members (Figure 6 and Supplementary File 8,9; revised manuscript line #197-210).

4) Figure 1b, what about expression levels of other Obox genes in TBLCs vs. ESC?

We thank the reviewer for raising this point. We sought to identify ZGA drivers by looking for homeobox genes that are expressed during ZGA. In the Obox family, only *Obox1/2/3/4/5/*7 meet this criterion. Because *Obox6* and *Obox8* are not expressed during ZGA, they are not shown in the figures. Since the original TBLC data lacks replicate and the repetitiveness of *Obox* family members, transcripts with TPM less than 1.0 were further discarded in our analysis pipeline, leaving only *Obox4*. We have re-analyzed the TBLC data by Shen *et al.* and expression levels of all *Obox* family members are shown below (Table R1).

**Author response table 1. sa2table1:** Expression levels of all *Obox* family members in mouse pluripotent stem cell (PSC) and totipotent blastomere-like cell (TBLC).

	PSC (TPM)	TBLC (TPM)	ZGA Expression
Oboxl Obox2Obox3Obox4Obox5Obox6Obox7Obox8	0.000.000.000.090.000.780.000.90	0.000.000.262.090.117.240.001.02	TRUETRUETRUETRUETRUEFALSETRUEFALSE

5) Line 79, Obox4 should be partially affected by Dux KO based on Figure 1c and the legend.

We thank the reviewer for pointing this out. We have revised the statement accordingly.

6) Figure S1b, it seems that Dux is not expressed in overexpressed cells. Is this due to that Dux is quickly degraded?

Interestingly, we have been consistently observing that DUX appears as a 130 kDa band in western blot as it is shown in the figure, likely due to dimerization despite strong reducing and denaturing condition of the assay. The reason for a weak DUX band is likely synthetic. Degradation is one possibility. Another possibility is an enrichment of non-transfected cells in the sample due to a high cytotoxicity of *Dux*. Also, when transferring proteins from PAGE gels to membranes, larger proteins tend to be under-transferred.

7) Figure 2, How similar or different for Obox4 OE vs. Obox3 and 5 OE as reported by Ji et al? Are MT2C_Mm and MT2B2 also induced upon Obox4 OE?

We thank the reviewer for the suggestion of comparing overexpression of *Obox4* vs. other *Obox* family members. Answering this question would help to distinguish the targets of *Obox4* from its family and clarify its unique functional redundancy with *Dux*. By comparing *Obox3*/*Obox5* (representing zygotic and maternal *Obox*, respectively) induced genes reported by Ji *et al.* with our data, we found that while genes induced by *Obox3* and *Obox5* exhibit a strong overlap, the majority of their targets cannot be induced by *Obox4* (Author response image 3a). Per-gene expression analysis showed very low correlation between induction targets of *Obox3*/*Obox5* and *Obox4* (Author response image 3b). This suggests that *Obox4* has unique targeting pattern that is divergent from other *Obox* members. We have included this analysis in the revised manuscript (Figure 6A-D, revised manuscript line #200-206). Similar with genes, transposable elements targeted by *Obox3*/*Obox5* (e.g. MT2C_Mm and MT2B2) were not induced upon *Obox4* overexpression. Due to space constraints, the results of differential expression analysis of all transposable elements are present in Supplementary File 2 instead of the main figure.

**Author response image 2. sa2fig2:** Comparison of *Obox3*/*Obox4*/*Obox5* induced transcriptome. a) Venn diagram showing overlap of 2C genes induced by ectopic expression of Obox3, Obox4, and Obox5 in mESCs. b) Scatterplot showing per-gene expression changes in Obox3 versus Obox5 induced mESCs. 2C-genes are highlighted in red. c) Scatterplot showing per-gene expression changes in Obox3 (left panel) and Obox5 (right panel) induced versus Obox4 induced mESCs. 2C-genes are highlighted in red.

8) Figure 2h, does Dux also induce other Obox genes in addition to Obox4?

This is a very interesting point raised by the reviewer. Using data from Hendrickson *et al.*, our analysis showed that *Dux* overexpression in mESCs strongly induces *Obox1*/*2*/*3*/*6* (Supplementary File 2, also revised Figure 2H). This contradicts the analysis using the same data by Ji *et al.* that concludes “Obox genes, apart from Obox4, were not or only moderately affected by Dux overexpression in mES cells^16^ and Dux knockout in embryos^20^ (Extended Data Figure 10c,d)” (DoI: doi.org/10.1038/s41586-023-06428-3). In our own mESC RNA-seq (unpublished), we also observed strong induction of *Obox1*/*2*/*3*/*6* by *Dux* overexpression. Excluding multiple mapping reads from the analysis does not affect the results.

9) Figure S3b. Why much fewer peaks (~10%) of Dux CUT&RUN peaks vs. ChIP-seq peaks? Are the missing peaks false negatives?

We understand the reviewer’s concern on the coverage of the DUX CUT&RUN data. Here are a few considerations about the cause of fewer peaks compared with the previous ChIP-seq data:

- First, because our DUX CUT&RUN served as a proof-of-principle of profiling homeobox DNA binding by CUT&RUN with an anti-FLAG antibody, only a snapshot would be enough for de novo consensus motif discovery. Hence, we decided to run our DUX CUT&RUN at a sequencing depth significantly shallower than the previously reported ChIP by Hendrickson *et al.* (~11.3M read pairs vs. ~48.1M read pairs + 48.6M read pairs). Fewer peaks in accordance with fewer reads is within our expectation, as read depth significantly affects discover power in DNA binding profiling experiments (DoI: 10.1371/journal.pcbi.1003326).

- Second, as a proof-of-principle, we did not include biological/technical replicate in our DUX CUT&RUN experiment. This further lowered discover power.

- Third, we performed our CUT&RUN experiments using two different controls: (i) a non-immune mouse IgG in *Obox4* induced mESCs, and (ii) anti-FLAG antibody in WT mESCs. The DUX CUT&RUN data were normalized to both controls for increased peak calling stringency, which lead to reduced number of peaks called.

10) Obox1/3/5 seem also bind to B1/B2 elements in addition to ERVL. How about Obox4? Since ESCs chromatin structure is very different from 2-cell embryos. The authors may consider profiling Obox4 binding in embryos.

Our CUT&RUN data showed that OBOX4 barely binds B1/B2 elements, at least in mESCs. True, the 2C embryo represents a unique stage where chromatin is highly accessible, and thus profiling 2C embryos would theoretically better capture the DNA binding property of OBOX4. Regretfully, in our practice, the single cell/low input profiling technique (scCUT&Tag) does not generate enough high-quality data (data not shown). We have included a table containing OBOX4 coverage of transposable element loci by family (Supplementary Table 5).

11) Since one Obox4 copy with ORF was not depleted according to the KO design, the authors should perform immunostaining to confirm whether Obox4 protein is gone in Obox4 KO embryos.

We understand the reviewers concern and agree that the percentage of loci left does not necessarily correlate with the overall expression. However, considering the complex interplay between *Dux* and *Obox4* where the two factors express in a mutually inductive fashion, the depletion of OBOX4 should be examined in a concomitant DUX depletion setting. Unfortunately, because DKO mating pairs are unavailable, we had to focus on the characterization of our DKD model. As a reference, our data showed that OBOX4 protein cannot be detected by immunostaining under ASO knockdown, where the *Obox4* transcript level was reduced to roughly 15% of that in scramble ASO injected embryos.

12) Figure 4i is confusing, why very few Dux-Het/Obox4KO embryos reach to blastocyst at 4.5 dpc? In theory, Dux-Het/Obox4KO should show no embryonic arrest based on Figure 4e-g.

This is a very interesting point raised by the reviewer. We think this is related to the reviewer’s comment 1 where DUX/OBOX4 depletion caused more severe phenotype in vitro than in vivo. As mentioned in our response to reviewer’s comment 1, in vitro culturing represents a major challenge to preimplantation embryogenesis, particularly to blastulation. Our theory is that while dosage effect of mono-allelic *Dux* in the Dux-Het/Obox4KO embryos can support ZGA to some extent to allow development beyond 2C stage, the ZGA is too crippled to overcome the adversary of in vitro culture.

13) Since Obox4 has a high identity to other Obox proteins (Figure S11), authors should demonstrate the antibody specificity.

As shown in Figure 1—figure supplement 1B,D: OBOX4 has the lowest amino acid sequence homology to other OBOX family members. In Figure 1—figure supplement 1C, the specificity of our monoclonal antibodies was extensively tested against OBOX2, the closest though still distal, paralog of OBOX4. The specificity is further demonstrated by the KD experiment, where ASO targeting a unique sequence of *Obox4* was used (Figure 4—figure supplement 5A-C).

14) Line 231-233. "However, whether and how other Obox family members contribute to ZGA remains unclear". This is not true, because Ji et al. has showed that Obox is critical in regulating ZGA.

The sentence is removed from the revised Discussion section.

15) Line 56-57. "Nr5a2 that triggers ZGA by activating short interspersed nuclear element (SINE) B1 family". The authors should be cautious about this statement because it has been shown that Nr5a2 had a minimal role in ZGA by maternal-zygotic KO mouse model (https://doi.org/10.1101/2023.01.16.524255).

We really appreciate the reviewer’s reminder. This is consistent with the recent publication by Lai *et al.* who demonstrated *Nr5a2* knockout is compatible with ZGA (DoI: 10.1038/s41422-023-00887-z). The original remark is removed from the revised Discussion section.

16) I feel that only a few representative genotyping gel images are sufficient. The authors do not need to show all the raw genotyping gels. In addition, Figure 5b (A+B) gel is noisy and it's difficult to locate the target bands.

We thank the reviewer’s considerate remark. We have adjusted the figure to increased readability. Considering the complexity of the knockout experiments involving two multi-copy tandem repeat gene clusters, we would like to keep the complete array of genotyping image in case some of the readers are curious about the experimental design.

17) Typo in Figure S3a, it should be Dux instead of Obox?

Thanks for the correction, the typo is fixed.